# ON THE NECESSITY OF DISENTANGLED REPRESENTATIONS FOR DOWNSTREAM TASKS

## ABSTRACT

A disentangled representation encodes generative factors of data in a separable and compact pattern. Thus it is widely believed that such a representation format benefits downstream tasks. In this paper, we challenge the necessity of disentangled representation in downstream applications. Specifically, we show that dimension-wise disentangled representations are not necessary for downstream tasks using neural networks that take learned representations as input. We provide extensive empirical evidence against the necessity of disentanglement, covering multiple datasets, representation learning methods, and downstream network architectures. Moreover, our study reveals that informativeness of representations best accounts for downstream performance. The positive correlation between the informativeness and disentanglement explains the claimed usefulness of disentangled representations in previous works.

## 1 INTRODUCTION

Disentanglement has been considered an essential property of representation learning (Bengio et al., 2013; Peters et al., 2017; Goodfellow et al., 2016; Bengio et al., 2007; Schmidhuber, 1992; Lake et al., 2017; Tschannen et al., 2018). Though there is no widely accepted formal definition yet, the fundamental intuition is that a disentangled representation should separately and distinctly capture information from generative data factors (Bengio et al., 2013). In practice, disentanglement is often implemented to emphasize a *dimension-wise* relationship, i.e., a representation dimension should capture information from exactly one factor and vice versa (Locatello et al., 2019b; Higgins et al., 2016; Kim & Mnih, 2018; Chen et al., 2018; Eastwood & Williams, 2018; Ridgeway & Mozer, 2018; Kumar et al., 2017; Do & Tran, 2019). Disentangled representations offer human-interpretable factor dependencies. Therefore, in theory, they are robust to variations in the natural data and are expected to benefit downstream performances (Bengio et al., 2013).

Researchers are interested in empirically verifying these purported advantages. Especially, they focus on the following two-staged tasks: **(1)** extracting representations in an unsupervised manner from data, **(2)** then performing downstream neural networks training based on learned representations (van Steenkiste et al., 2019; Locatello et al., 2019a; Dittadi et al., 2020; Locatello et al., 2020). Among various downstream tasks, except the ones that explicitly require disentanglement (Higgins et al., 2018b; Gabbay & Hoshen, 2021; Schölkopf et al., 2021), abstract visual reasoning is widely recognized as a popular testbed (van Steenkiste et al., 2019; Locatello et al., 2020; Schölkopf et al., 2021). The premise behind it aligns with the goals of machine intelligence (Snow et al., 1984; Carpenter et al., 1990). Moreover, its mechanism ensures valid measurement of representations downstream performance (Fleuret et al., 2011; Barrett et al., 2018).

In the abstract visual reasoning task, intelligent agents are asked to take human IQ tests, i.e., predict the missing panel of Raven's Progressive Matrices (RPMs) (Raven, 1941). Indeed it is a challenging task for representation learning (Barrett et al., 2018; van Steenkiste et al., 2019). Disentanglement literature often takes this task as an encouraging example to show that disentanglement leads to quicker learning and better final performance (van Steenkiste et al., 2019; Locatello et al., 2020; Schölkopf et al., 2021).

However, on the abstract visual reasoning task, we find that rotating disentangled representations, i.e., multiplying the representations by an orthonormal matrix, has *no* impact on sample efficiency and final accuracy. We construct the most disentangled representations, i.e., normalized true factors.

Then we solve the downstream tasks from them and their rotated variants. As shown in Figure 2a, there is little difference between the accuracy curves of original and rotated representations throughout the learning process. On one hand, this phenomenon is surprising since the rotation decreases dimension-wise disentanglement by destroying axis alignment (Locatello et al., 2019b). Indeed, in Figure 2b we can observe notable drops in disentanglement metric scores (first 5 columns). Our finding demonstrates that disentanglement does not affect the downstream learning trajectory, which is against the commonly believed usefulness of disentanglement. On the other hand, it is not surprising since we apply an invertible linear transform. We can observe that *Logistic Regression* (LR) accuracy remains 100% before and after rotation, indicating that a simple linear layer could eliminate the effects of rotation.

Per such facts, some questions arise: *Are disentangled representations necessary for two-staged tasks?* If not, *which property matters?* To address them, we conduct an extensive empirical study based on abstract reasoning tasks. Our contributions are as follows.

- We challenge the necessity of disentanglement for abstract reasoning tasks. We find that **(1)** entangling representations by random rotation has little impact, and **(2)** general-purpose representation learning methods could reach better or competitive performance than disentanglement methods.

- Following Eastwood & Williams (2018), we term *what* information the representation has learned as *informativeness*. We show that *informativeness* matters downstream performance most. **(1)** *Logistic regression* (LR) accuracy on factor classification correlates most with downstream performance, comparing with disentanglement metrics. **(2)** Conditioning on close LR accuracy, disentanglement still correlates mildly. **(3)** The informativeness is behind the previously argued usefulness of disentanglement since we observe a positive correlation between LR and disentanglement metrics.

- We conduct a large-scale empirical study supporting our claim. We train 720 representation learning models covering two datasets, including both disentanglement and general-purpose methods. Then we train 5 WReNs (Barrett et al., 2018) and 5 Transformers (Vaswani et al., 2017; Hahne et al., 2019) using the outputs of each representation learning model to perform abstract reasoning, yielding a total of 7200 abstract reasoning models.

## 2 RELATED WORK

**Disentangled representation learning.** There is no agreed-upon formal definition of disentanglement. Therefore, in practice, disentanglement is often interpreted as a one-to-one mapping between representation dimensions and generative factors of data, which we term "dimension-wise disentanglement". It requires that the representation dimension encode only one factor and vice versa (Locatello et al., 2019b; Eastwood & Williams, 2018; Kumar et al., 2017; Do & Tran, 2019). Besides dimension-wise disentanglement, Higgins et al. (2018a) propose a definition from the group theory perspective. However, its requirement in interaction with the environment prevents applicable learning methods for existing disentanglement benchmarks (Caselles-Dupré et al., 2019).

Adopting the dimension-wise definition, researchers develop methods and metrics. SOTA disentanglement methods are mainly variants of generative methods (Higgins et al., 2016; Kim & Mnih, 2018; Burgess et al., 2018; Kumar et al., 2017; Chen et al., 2018; 2016; Jeon et al., 2018; Lin et al., 2020). Corresponding metrics are designed in the following ways (Zaidi et al., 2020): intervening factors (Higgins et al., 2016; Kim & Mnih, 2018), estimating mutual information (Chen et al., 2018), and developing classifiers (Eastwood & Williams, 2018; Kumar et al., 2017). Another line of work related to disentangled representation learning is the Independent Component Analysis (ICA) (Comon, 1994). ICA aims to recover independent components of the data, using the mean correlation coefficient (MCC) as the metric. However, ICA models require access to auxiliary variables (Hyvarinen et al., 2019), leading to inevitable supervision for image datasets training (Hyvarinen & Morioka, 2016; Khemakhem et al., 2020a;b; Klindt et al., 2020). In this paper, we focus on the downstream performance of unsupervised representation learning.

**Downstream tasks.** It is widely believed that disentangled representations benefit downstream tasks. Intuitively, they offer a human-understandable structure with ready access to salient factors, hence should be enjoying robust generalization capacity (Bengio et al., 2013; Do & Tran, 2019). Several works conduct empirical studies on downstream tasks to support the notions above, includ-

ing abstract reasoning (van Steenkiste et al., 2019), fairness (Locatello et al., 2019a), and sim2real transfer (Dittadi et al., 2020). Among these works, van Steenkiste et al. (2019) provide the most encouraging evidence from abstract reasoning tasks. We adopt their settings and investigate the same tasks. However, their results are questionable. Firstly, it underestimates factors' linear classification accuracy, yielding a weaker correlation between informativeness and downstream performance (see Figure 9 in Appendix A.3). Moreover, only variants of VAEs are considered. We address these issues and achieve opposite conclusions.

**Abstract visual reasoning** has been a popular benchmark to measure the representation's downstream performance, especially in disentanglement literature (Steenbrugge et al., 2018; van Steenkiste et al., 2019; Dittadi et al., 2020; Locatello et al., 2020; Schölkopf et al., 2021). The most common type is the Raven's Progressive Matrices (RPMs) (Raven, 1941), which highly emphasize abstract and relational reasoning capacities and effectively represent human intelligence (Snow et al., 1984; Carpenter et al., 1990). To solve RPMs, one is asked to complete the missing panel of a $3 \times 3$ grid by exploring the logical relationships of 8 context panels. Moreover, abstract visual reasoning is a well-developed benchmark for representation learning. Given that it is coupled with a principle treatment of generalization (Fleuret et al., 2011), a neural network can not solve reasoning tasks by simply memorizing superficial statistical features. Besides, it can avoid pitfalls where test-specific heuristics learned by downstream models obscures the original properties of representations (Barrett et al., 2018). To summarize, **(1)** the goal of abstract visual reasoning highlights our requirements for representation learning, and **(2)** its mechanism ensures valid measurements. For these reasons, we focus on the necessity of disentanglement for the abstract reasoning task.

## 3 DOWNSTREAM BENCHMARK: ABSTRACT VISUAL REASONING

This section contains background on the downstream benchmark framework. We first introduce the definition of the abstract visual reasoning task. Then we present the framework's ingredients: representation learning methods, metrics, and abstract reasoning models.

### 3.1 ABSTRACT VISUAL REASONING AS A TWO-STAGED TASK

The abstract visual reasoning tasks are highly inspired by the famous human IQ test, Raven's Progressive Matrices (RPMs) (Raven, 1941). Figure 1 shows an RPM question in our evaluation dataset. There are eight context panels and one missing panel in the left part of the figure. The context panels are arranged following some logical rules across rows. During the test, the subject must pick one of the six candidates listed in the right part to fix the missing panel. The goal is to maintain the logical relationships given by the contexts. More details of RPMs are available in Appendix A.4.

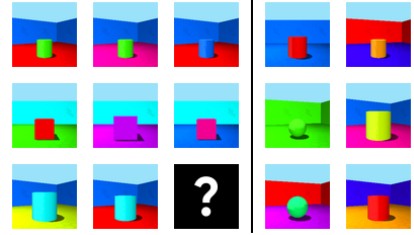

Figure 1: An example of RPM on *3DShapes* from van Steenkiste et al. (2019).

We adopt RPMs as a downstream benchmark following van Steenkiste et al. (2019). To measure the necessity of disentanglement for downstream tasks, we separate the evaluation process into two stages: **(1)** In Stage-1, representation learning models extract representations from images of which RPMs consist, and **(2)** in Stage-2, abstract reasoning models predict the missing panels from the frozen representations of contexts and answer candidates. Correspondingly, we denote representation learning models as **Stage-1 models** while abstract reasoning models as **Stage-2 models**. For Stage-1, we measure the disentanglement properties of the representations. A diverse set of Stage-1 and Stage-2 models are trained, yielding multiple samples from the joint distribution of representation metric scores and downstream accuracy. Finally, we study the relationships between representation qualities and downstream performance. We aim to investigate whether more disentangled representations perform better on abstract reasoning tasks.

The two-staged framework leverages large-scale experiments to reveal connections between the disentanglement of representations and their downstream performance. It provides a precise measurement of the importance of disentanglement. Therefore the two-staged framework is widely-accepted (van Steenkiste et al., 2019; Locatello et al., 2019a; Dittadi et al., 2020; Locatello et al., 2020).

### 3.2 BACKGROUND OF REPRESENTATION LEARNING

**Disentangled representation learning methods.** The seminal works of Higgins et al. (2016) and Chen et al. (2016) embody disentanglement by augmenting deep generative models (Kingma & Welling, 2013; Goodfellow et al., 2014). For disentangled representation learning methods, we focus on variants of VAE. Namely, $\beta$-VAE (Higgins et al., 2016), AnnealedVAE (Burgess et al., 2018), $\beta$-TCVAE (Chen et al., 2018), FactorVAE (Kim & Mnih, 2018), and DIP-VAE (Kumar et al., 2017). They achieve disentanglement mainly by encouraging independence between representation dimensions. Please refer to Appendix A.2 for details.

**General-purpose representation learning methods.** In our study, methods not (explicitly) encouraging disentanglement are called general-purpose methods. We take BYOL (Grill et al., 2020) as a representative. BYOL is a negative-free contrastive learning method. It creates different "views" of an image by data augmentation and pulls together their distance in representation space. To avoid collapsing to trivial representations, a predictor appending to one of the siamese encoders and exponential moving average update strategy (He et al., 2020) are employed. It does not encourage disentanglement due to the lack of regularizers. Indeed, the empirical evidence in Cao et al. (2022) demonstrates that representations learned by BYOL have weak disentanglement.

**Representation property metrics.** Considered properties of representations cover two axes of metrics: disentanglement metrics and informativeness metrics (Eastwood & Williams, 2018). We include *BetaVAE* score (Higgins et al., 2016), *FactorVAE* score (Kim & Mnih, 2018), *Mutual Information Gap* (Chen et al., 2018) , *SAP* (Kumar et al., 2017), and *DCI Disentanglement* (Eastwood & Williams, 2018). Locatello et al. (2019b) proves their agreement on VAE methods with extensive experiments. Though their measurements are different, their results are positively correlated. On the other hand, informativeness requires representations to encode enough information about factors. In this work, we employ *Logistic Regression* (LR). It is a favorable metric adopted by unsupervised pretraining literature (He et al., 2020; Grill et al., 2020; Caron et al., 2021). Given the weak capacity of linear models, a higher LR accuracy ensures that sufficient information is explicitly encoded. However, it does not emphasize a dimension-wise encoding pattern like disentanglement. To distinguish, we term the property indicated by LR as *informativeness*.

### 3.3 BACKGROUND OF METHODS FOR ABSTRACT REASONING

In Stage-1, we extract representations of eight context panels (the left part of Figure 1) and six answer candidates (the right part of Figure 1). Then in Stage-2, downstream models perform abstract reasoning from the (frozen) representations. Abstract reasoning models evaluate whether filling the blank panel by a candidate follows the logical rules given by contexts. For a trial $T_i$ of one candidate $a_i \in A = \{a_1, ..., a_6\}$ and eight context panels $C = \{c_1, ..., c_8\}$, its score is calculated as follows:

$$Y_i = \text{Stage2}(\text{Stage1}(T_i)), \quad \text{Stage1}(T_i) = \{\text{Stage1}(c_1), \ldots, \text{Stage1}(c_8)\} \cup \{\text{Stage1}(a_i)\}, \quad (1)$$

where $Y_i$ is the score of trial $T_i$, $\text{Stage1}(\cdot)$, $\text{Stage2}(\cdot)$ denote the forward process of the Stage-1 and Stage-2 models, and $\text{Stage1}(T_i)$ is the representations of contexts and candidate $a_i$. After evaluating all trials $\{T_1, T_2, \ldots, T_6\}$, the output answer is $\arg\max_i Y_i$.

We implement two different well-defined structures of Stage-2 models, namely, WReN (Barrett et al., 2018) and Transformer (Vaswani et al., 2017; Hahne et al., 2019). First, they employ an MLP or a Transformer to embed an RPM trial. Then an MLP head predicts a scalar score from the embeddings.

## 4 EXPERIMENTS

In this Section, we conduct a systematic empirical study about representation properties' impacts on downstream performance. First, we introduce our experimental conditions in Section 4.1. Then we provide empirical evidence to challenge the necessity of disentanglement (Section 4.2) and to tell which property matters (Section 4.3).

### 4.1 EXPERIMENTS SETUP

We build upon the experiment conditions of van Steenkiste et al. (2019). Abstract visual reasoning tasks, i.e., RPMs, are solved through a two-stage process: data $\xrightarrow{\text{Stage-1}}$ representations $\xrightarrow{\text{Stage-2}}$

RPM answers. We first train Stage-1 models in an unsupervised manner and evaluate their disentanglement and informativeness. Then Stage-2 models are trained and evaluated on downstream tasks, yielding an abstract reasoning accuracy of a representation. Provided with a large amount of (representation property score, downstream performance) pairs, we conduct a systematic study to investigate the necessity of disentanglement. More implementation details are available in Appendix A.

**Datasets.** We replicate the RPM generation protocol in van Steenkiste et al. (2019). The panel images consist of disentanglement benchmark image datasets, namely, *Abstract dSprites* (Matthey et al., 2017; van Steenkiste et al., 2019) and *3DShapes* (Burgess & Kim, 2018). The rows of RPMs are arranged following the logical *AND* of ground truth factors. As for hardness, we only reserve *hard-mixed*, whose contexts and candidates are more confusing. According to the generation process, the size of generated RPMs is sufficiently large (about $10^{144}$), allowing us to produce fresh samples throughout training.

**Reference models.** Stage-1 models extract representations from RPM's panels. To ensure the generalizability of the results, we include 360 disentangled VAEs (denoted as DisVAEs) and 360 BYOLs. Our choices of Stage-1 models cover both disentangled and general-purpose representation learning methods. Moreover, we are interested in the *overall* relationship between representation properties and downstream performance. Therefore we need to study the correlation between two distributions, i.e., representation metric scores and downstream performance. For this, we include various samples for both Stage-1 and Stage-2 to ensure they are representative enough. For Stage-1, a diverse set of configurations are included for each type of representation learning model. According to the histograms in Appendix C.4, our choices span various disentanglement and informativeness scores. For Stage-2, to better estimate the downstream performance distribution, we use multiple Stage-2 configurations for each representation instead of searching for the best one. Specifically, we train 10 Stage-2 models (5 WReNs and 5 Transformers) for every Stage-1 model. Stage-2 configurations are randomly sampled from a search space described in Appendix A.3 and shared across Stage-1 models. By this, we ensure fair comparisons across representations.

**Training protocol.** Training is conducted two-staged. Firstly, we train Stage-1 models in an unsupervised manner on the dataset consisting of RPMs' panels, i.e., *Abstract dSprites* or *3DShapes*. For DisVAE models, we use the training protocol of van Steenkiste et al. (2019), while for BYOL models, we follow Cao et al. (2022). In Stage-2, all models are trained for 10K iterations with a batch size of 32. After every 100 iterations, we evaluate the accuracy on newly generated 50 mini-batches of unseen RPM samples for validation and another 50 mini-batches for testing.

**Evaluation protocol.** We first evaluate the two stages separately. Then we analyze the relationship between the two stages, i.e., representation properties and downstream performance. Specifically, to challenge the necessity of disentanglement, we are interested in whether more disentangled representations lead to better downstream performance. Further, if it turns out that disentanglement is of limited importance, can we find another metric that better accounts for downstream performance? Therefore, for Stage-1, we employ representation metrics described in Section 3.2 to measure two aspects: disentanglement and informativeness. For all Stage-1 models, we compute the following metric scores: *BetaVAE* score, *FactorVAE* score, *MIG*, *SAP*, and *LR* accuracy. *DCI Disentanglement* is only evaluated for DisVAEs since it takes hours to develop the Gradient Boosting Trees required during the evaluation process on high-dimensional representations of BYOLs (Cao et al., 2022). For Stage-2, we inspect accuracy on newly generated test sets every 100 iterations, yielding accuracy for multiple training steps. Since every step sees fresh samples, we employ training curves to measure sample efficiency. We also report accuracy-#samples curves in Appendix C.2 .

To summarize the downstream performance of a Stage-1 model, over 5 WReNs or 5 Transformers in Stage-2, we report the mean accuracy denoted as $\overline{\text{WReN}}$ or $\overline{\text{Trans.}}$, and max accuracy denoted as WReN$^\star$ or Trans.$^\star$. Finally, we calculate the rank correlation (Spearman) between the mean performance of Stage-1 models ($\overline{\text{WReN}}$ and $\overline{\text{Trans.}}$) at certain Stage-2 steps and their Stage-1 metric scores. A larger correlation indicates a higher significance of the representation property on downstream performance.

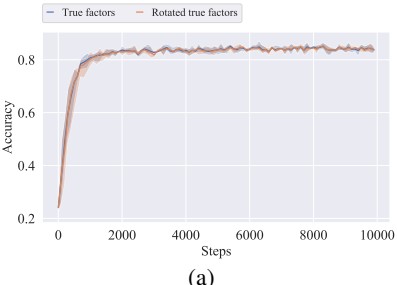
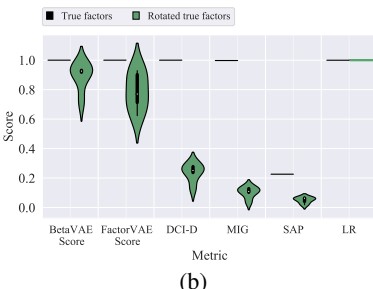

(a)                                    (b)

Figure 2: Downstream accuracy and metric scores of original/rotated true factor values with 5 configurations of rotation matrices and downstream model structures. (a) Average downstream accuracy throughout training. The shaded area indicates max and min values. (b) The violin plot of metric scores of original/rotated true factor values. Scores with deterministic values collapse to a line. The first 5 metrics measure disentanglement while LR measures informativeness.

## 4.2    ARE DISENTANGLED REPRESENTATIONS NECESSARY?

Hereafter we challenge the necessity of disentanglement. We begin by comparing a disentangled representation v.s. a deliberately designed, entangled representation on the downstream performance. Then we discuss the necessity of disentanglement inductive bias by evaluating the performance of disentanglement and general-purpose representation learning methods.

**Effects of attenuating disentanglement.** We first construct the most disentangled representations, i.e., the normalized true factor values. We normalize the true factor values to have zero means and unit standard deviations, yielding 6-d representations (note that *Abstract dSprites* and *3DShapes* are both labeled with 6 ground truth factors). Then we rotate the constructed representations by multiplying randomly generated orthonormal matrices. Afterward, each dimension of the rotated feature captures a combination of factors, thus destroying disentanglement. Finally, we perform abstract reasoning training from true factors before and after rotations. We also conduct rotations on representations learned by DisVAEs.

We run 5 seeds defining the randomly generated rotation matrices and Stage-2 model configurations. We report results on *3DShapes* with original/rotated true factors as representations and WReNs as Stage-2 models in Figure 2. As depicted in Figure 2a, there is little difference between performance before and after rotation throughout the training process. Yet Figure 2b shows significant drops in disentanglement metric scores. This surprising phenomenon suggests that even though we drastically entangle the representations, the downstream performance remains unchanged, firmly against the necessity of disentanglement. However, we can see from Figure 2b that LR scores are 100% before and after rotation. It is easy to understand because the rotation we applied

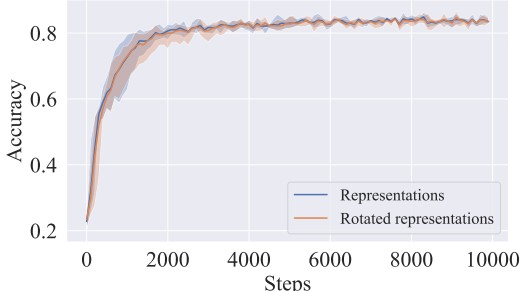

Figure 3: Average downstream accuracy of DisVAEs' representations w/ and w/o rotation on *3DShapes* with WReN as Stage-2 models. We select the most disentangled DisVAE in terms of *FactorVAE* score. The two curves are almost identical.

is just an invertible linear transform, which a simple LR can recover, not to mention more capable Stage-2 models. Moreover, we observe similar results for learned representations (Figure 3). We select the most disentangled DisVAE measured by *FactorVAE* score among the 180 DisVAE models trained on *3DShapes* (recall Section 4.1). As shown in Figure 3, rotation does not hurt the performance of representations learned by DisVAEs, backing up our claim that disentanglement representations might not be necessary to achieve good downstream performance. More results of rotation experiments on other datasets are reported in Appendix C.3.

**Summary:** Destroying disentanglement (by random rotation) in representations does not have a noticeable impact on downstream performance throughout training.

**Advantages of disentanglement inductive bias.** From previous results, we demonstrate that both high performance and high sample efficiency can be achieved even if we deliberately destroy disen-

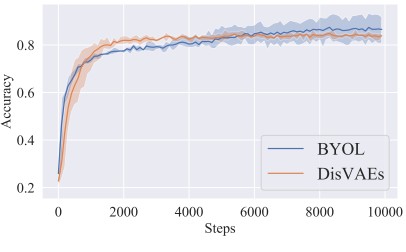
(a) Stage-2=WReN

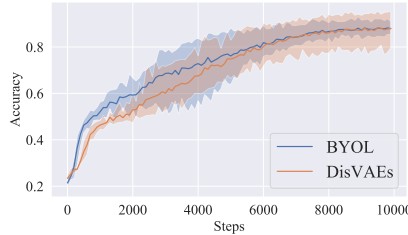
(b) Stage-2=Transformer

Figure 4: Average test accuracy on *3DShapes* throughout the training. We select the Stage-1 models with best $\overline{\text{WReN}}$ or $\overline{\text{Trans.}}$ among 3600 checkpoints on *3DShapes*. Stage-1 models with disentanglement inductive bias (DisVAEs) are not necessarily better than those without such bias (BYOL) in terms of sample efficiency and final accuracy.

Table 1: Best downstream test performance (%) of different stage1 models. The step with the highest validation accuracy is reported. The numbers in the parentheses are STDs.

| Dataset | Stage1 | WReN* | $\overline{\text{WReN}}$ | Trans.* | $\overline{\text{Trans.}}$ |
|---|---|---|---|---|---|
| *3DShapes* | DisVAEs | 89.9 | 84.8(0.91) | **96.4** | 87.0(6.36) |
| | BYOL | **93.9** | **87.1(4.68)** | 95.0 | **88.0(2.62)** |
| *Abstract dSprites* | DisVAEs | 76.5 | 68.7(1.39) | 72.5 | 66.4(7.06) |
| | BYOL | **78.3** | **72.2(3.11)** | **81.4** | **78.1(1.75)** |

tanglement. Further, we are interested in the inductive biases of Stage-1 models: Do disentangled representation learning models have absolute advantages on downstream performance over general-purpose models? For this, we compare the downstream performance of different families of learning models described in Section 4.1, including BYOL, $\beta$-VAE, AnnealedVAE, $\beta$-TCVAE, FactorVAE, DIP-VAE-I, and DIP-VAE-II. Among them, BYOL does not explicitly encourage disentanglement. On the other hand, all DisVAEs are disentangled representation learning methods. From a large pool of 7200 checkpoints, we report the best performance for each model family.

Figure 4 shows overviews of training trajectories of Stage-1 models with the highest performing $\overline{\text{WReN}}$ and $\overline{\text{Trans.}}$ on *3DShpaes* for multiple training steps. For WReN as Stage-2 models (Figure 4a), BYOL leads at the beginning, then DisVAEs catch up, and finally, BYOL converges at a higher accuracy. In contrast, when Stage-2 models are Transformers, BYOL's curve grows faster, but DisVAEs and BYOL converge with comparable performance. In general, the two curves evolve in almost identical patterns with small gaps, indicating that disentanglement inductive bias is of limited utility in improving downstream sample efficiency. Corresponding analysis on *Abstract dSprites* is available in Appendix C.3, where we reach the same conclusions. As for final performance, we report maximal $\overline{\text{WReN}}$, WReN*, $\overline{\text{Trans.}}$ and Trans.* across different Stage-2 models and datasets in Table 1. We select checkpoints to evaluate based on validation accuracy. In particular, the best $\overline{\text{WReN}}$ and $\overline{\text{Trans.}}$ of BYOL are higher than that of DisVAEs'. In addition, it appears that BYOL performs better than or on par with DisVAEs in terms of WReN* and Trans.*. Especially, BYOL outperforms DisVAEs on *Abstract dSprites* with a considerable margin.

**Summary:** Models not intended for disentangled representation learning can reach superior or comparable downstream performance. Therefore disentanglement inductive bias does not necessarily lead to better sample efficiency or final accuracy.

### 4.3 WHICH PROPERTY MATTERS DOWNSTREAM PERFORMANCE?

The results in Section 4.2 provide encouraging cases against the necessity of disentanglement. Additionally, we are interested in several further issues: **(1)** Which property matters downstream performance most? **(2)** How can we interpret the previously claimed benefits from disentanglement(Bengio et al., 2013; Higgins et al., 2016; van Steenkiste et al., 2019; Locatello et al., 2019a; Dittadi et al., 2020)? On account of these questions, we start by investigating how different representation properties influence downstream accuracy. We include informativeness and various disentanglement metrics.

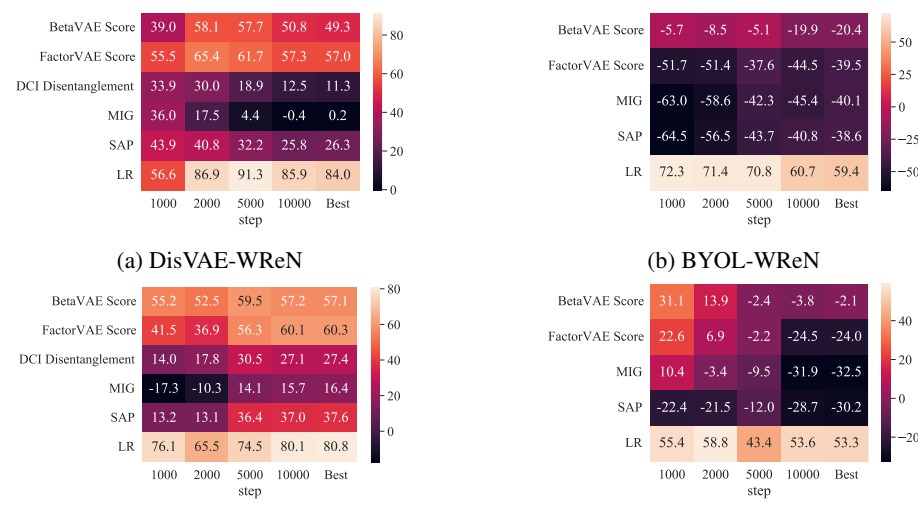

Figure 5: Rank correlations between $\overline{\text{WReN}}$ or $\overline{\text{Trans.}}$ and representation metrics on *3DShapes*. We denote the step with the highest validation accuracy as "Best". The brighter the panel, the more correlated the representation metric is with the downstream performance.

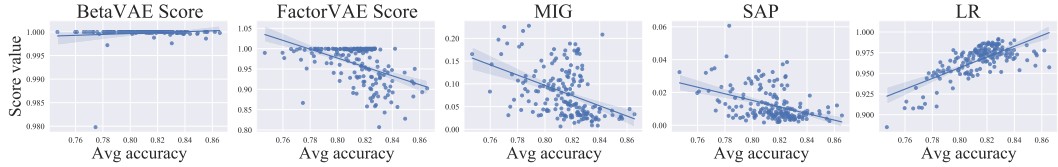

Figure 6: Representation metrics versus $\overline{\text{WReN}}$ at step 10000, where Stage-1 models are BYOL, and the dataset is *3DShapes*. We can observe a strong positive correlation between the informativeness metric scores and downstream accuracy.

Recall that we train 720 Stage-1 and 7200 Stage-2 models (see Section 4.1). By taking $\overline{\text{WReN}}$ and $\overline{\text{Trans.}}$ as measurements (average reasoning accuracy over 5 WReNs or 5 Transformers), we yield 720 representations paired with their downstream performance. Generally, our analysis is based on rank correlation (Spearman) between representation metric scores and downstream performance. If the correlation score is high, we can conclude that the representation property measured by the considered metric score is significant to downstream performance.

**The representation property of the most significance.** We calculate the rank correlation between downstream accuracy and disentanglement and informativeness scores. Meanwhile, we report rank correlation at steps 1K, 2K, 5K, and 10K, and the step with the highest validation accuracy. From correlations at different training steps, we can tell how a representation property affects sample efficiency.

Figure 5 displays rank correlations between representation metric scores and abstract reasoning test accuracy on *3DShapes*. Firstly we can find that *Logistic Regression* accuracy (LR) correlates most with downstream performance. The strong correlation is exploited for all considered models at multiple steps. Since LR requires sufficient information to be captured and extracted easily from representations, we can conclude that the informativeness matters most in broad conditions. In contrast, we observe that the importance of disentanglement varies among Stage-1 model families. Disentangled representation learning models (DisVAEs) exhibit strong positive correlations for several disentanglement metrics (but weaker than LR), such as *FactorVAE* score and *DCI Disentanglement*. However, their significance does not apply to BYOL, where the correlation of disentanglement is mild or even negative. In Figure 6 we plot the ($\overline{\text{WReN}}$, metric score) pairs at step 10000. Indeed, for BYOL-WReN on 3DShapes, we can see the linear regression provides a good fit of downstream accuracy and informativeness metrics. As for disentanglement metrics, we can see that *BetaVAE* score and *FactorVAE* score suffer from narrow spreads. For *MIG* and *SAP*, the regression lines have negative slopes. We conduct a similar analysis on *Abstract dSprites* and take the same observations. Please refer to Appendix C.4 for more details.

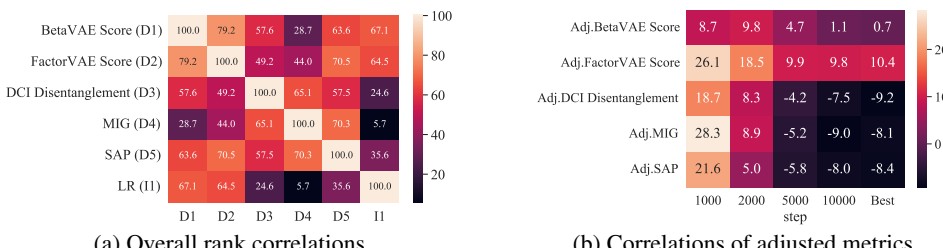

(a) Overall rank correlations.  (b) Correlations of adjusted metrics.

Figure 7: (a) Correlations between metrics and (b) correlations between adjusted metrics and downstream accuracy when using DisVAEs-WReN pipeline on *3DShapes*. Disentanglement metrics exhibit positive correlations with LR. Yet when conditioned on close informativeness, their adjusted versions show mild correlations.

**Summary:** The informativeness influences downstream performance most. The results are consistent across datasets and model structures.

**Understanding for the previously claimed success of disentanglement.** Previous works (van Steenkiste et al., 2019; Locatello et al., 2019a; Dittadi et al., 2020; Locatello et al., 2020) have reported empirical evidence backing up the advantages of disentangled representations. Consistently, we observe relatively strong correlations with disentanglement metrics, especially when Stage-1 models are DisVAEs in Figure 5. Based on our conclusion on the significance of the informativeness, we study the DisVAE-WReN case and provide some insights to explain why the disentanglement metrics have a high correlation to downstream performance in some cases.

We compute the overall correlations between metrics. The results are shown in Figure 7. For DisVAEs, we find that informativeness and disentanglement have high correlation scores. In particular, we can observe relatively strong correlations between LR and *FactorVAE* score and *BetaVAE* score. Accordingly, these disentanglement metrics exhibit relatively strong correlations with downstream performance in Figure 5a. In contrast, other disentanglement metrics correlate mildly with LR. And they are ineffective for downstream performance. Therefore, disentanglement metrics are not truly predictable for downstream performance, but LR is.

To "purify" the effect of disentanglement, a natural question is: If two representations are of close informativeness, does the more disentangled one more helpful for downstream tasks? For this, we employ adjusted metrics in Locatello et al. (2019a):

$$\text{Adj. Metric} = \text{Metric} - \frac{1}{5} \sum_{i \in N(\text{LR})} \text{Metric}_i, \qquad (2)$$

For a representation and a certain metric (we care more about disentanglement metrics), we denote its original metric score as $\texttt{Metric}$. Then we find its 5 nearest neighbors in terms of LR, which we write as $N(\text{LR})$. Finally, the difference between the original metric score and the mean score of the nearest neighbors is reported as adjusted metrics. Intuitively, we calculate the relative disentanglement for representations with close LR.

Figure 7b displays correlations between adjusted metrics and downstream performance. We can find that all adjusted disentanglement metrics correlate mildly with downstream performance. From this, we can see that when informativeness is close, being disentangled contributes only a small portion to the downstream performance when the downstream training steps are limited (In our case, less than or equal to 2000 steps, see Figure 4 and Figure 7).

**Summary:** The informativeness is the most predictable metric for downstream performance. Disentanglement only brings small extra benefits at the very beginning of downstream training.

## 5 CONCLUSION

In this paper, we challenge the necessity of dimension-wise disentanglement for downstream tasks. We conduct a large-scale empirical study on the abstract visual reasoning task. We start by showing that high downstream performance can be achieved by less disentangled representations. In addition, we identify that the informativeness is of the most significance. Finally, we conclude that dimension-wise disentanglement is unnecessary for downstream tasks using deep neural networks with learned representations as input.

## REPRODUCIBILITY STATEMENT

We provide information to reproduce our results in Appendix A. We commit to making our codes publicly available.

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

## A  REPRODUCIBILITY

In this Section, we provide implementation details to ensure reproducibility. In addition, we commit to making our codes, configurations, and running logs publicly available. All experiments are run on a machine with 2 Intel Xeon Gold 5218R 20-core processors and 4 Nvidia GeForce RTX 3090 GPUs.

### A.1  REPRESENTATION LEARNING METHODS

We include both disentangled representation learning methods and general-purpose representation learning methods. i.e., DisVAEs and BYOL (Grill et al., 2020).

**DisVAEs implementation.**  The DisVAEs include $\beta$-VAE (Higgins et al., 2016), Annealed-VAE (Burgess et al., 2018), $\beta$-TCVAE (Chen et al., 2018), FactorVAE (Kim & Mnih, 2018), and DIP-VAE-I and DIP-VAE-II (Kumar et al., 2017). We use the output of the encoder, the mean of $q_\phi(z|x)$, as representations. Hereafter, we introduce details for each method. The above methods encourage disentanglement by adding regularizers to ELBO. Adopting the notation in Tschannen et al. (2018), their objectives can be written in the following unified form:

$$\mathbb{E}_{p(x)}[\mathbb{E}_{q_\phi(z|x)}[-\log p_\theta(x|z)]] + \lambda_1 \mathbb{E}_{p(x)}[R_1(q_\phi(z|x))] + \lambda_2 R_2(q_\phi(z)), \quad (3)$$

where $q_\phi(z|x)$ is the posterior parameterized by the output of the encoder, $p_\theta(x|z)$ is induced by the decoder output, $R_1, R_2$ are the regularizer applying to the posterior and aggregate posterior, and $\lambda_1, \lambda_2$ are the coefficients controlling regularization. In the objective of $\beta$-VAE, $\beta = \lambda_1 > 1, \lambda_2 = 0$. Taking $R_1(q_\phi(z|x)) := D_{KL}[q_\phi(z|x)||p(z)]$ forces the posterior to be close to the prior (usually unit gaussian), hence penalizing the capacity of the information bottleneck and encourage disentanglement. FactorVAE and $\beta$-TCVAE takes $\lambda_1 = 0, \lambda_2 = 1$. With $R_2(q_\phi(z)) := TC(q_\phi(z))$, they penalize the Total Correlation (TC) (Watanabe, 1960). FactorVAE estimates TC by adversarial training, while $\beta$-TCVAE estimates TC by biased Monte Carlo sampling. Finally, DIP-VAE-I and DIP-VAE-II take $\lambda_1 = 0, \lambda_2 \geq 1$ and $R_2(q_\phi(z)) := ||\text{Cov}_{q_\phi(z)} - I||_F^2$, penalizing the distance between aggregated posterior and factorized prior.

We use the code and configurations from the DisLib [1] (Locatello et al., 2019b). As for parameters, we use the same sweep as van Steenkiste et al. (2019): for each one of the 6 DisVAEs, we use 6 configurations. We train each model using 5 different random seeds. Since we consider 2 datasets (*3DShapes* and *Abstract dSprites*), finally, we yield $6 * 6 * 5 * 2 = 360$ DisVAE checkpoints.

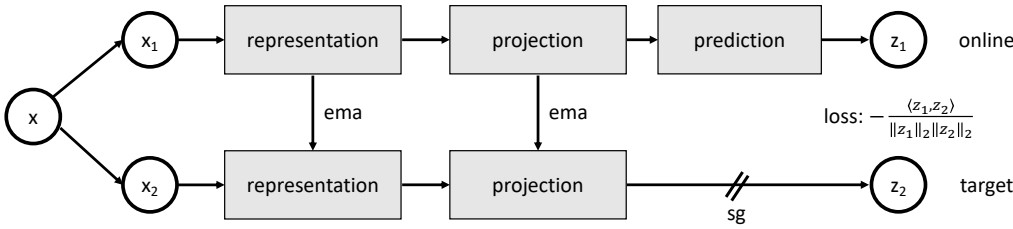

Figure 8: The pipeline of BYOL (Grill et al., 2020).

**BYOL implementation.**  BYOL (Grill et al., 2020) is a contrastive learning method. Figure 8 shows its pipeline. For each image $x$, we first create two "views" of it by data augmentation, i.e., $x_1$ and $x_2$. Then they are input to the siamese encoders: the online encoder and the target encoder. Specifically, $x_1$ is fed to the online encoder, while $x_2$ is fed to the target encoder, yielding the output

---
[1]https://github.com/google-research/disentanglement_lib.git

Table 2: The representation network architecture of our BYOL implementation, following Cao et al. (2022). Besides, there is a ReLU activation layer and a possible normalization layer following each convolutional layer to create a stack of (Conv-ReLU-Norm) blocks. The normalization stratege $norm$ and representation dimension $D$ are parameters to be set.

| **Representation Network** |
| --- |
| **input**: $64 \times 64$ images |
| **pipeline**: |
| $\quad$ 4×4 conv, stride 2, 32-channel |
| $\quad$ 4×4 conv, stride 2, 32-channel |
| $\quad$ 4×4 conv, stride 2, 64-channel |
| $\quad$ 4×4 conv, stride 2, 64-channel |
| $\quad$ 4×4 conv, stride 2, 128-channel |
| $\quad$ 1×1 conv, stride 1, $D$-channel |

$z_1$ and $z_2$, respectively. As for architectures, both encoders share the same representation network and projection MLP. The prediction MLP is appended to the online encoder in order to avoid BYOL learning trivial representations. The objective of BYOL is

$$\mathcal{L} = -\frac{\langle z_1, z_2 \rangle}{\|z_1\|_2 \|z_2\|_2}. \tag{4}$$

We are pulling the representations of the two "views" close. While training, the online encoder's parameters are updated by gradient descent. However, the target encoder's parameters are updated by the online parameters' Exponential Moving Average (EMA) (He et al., 2020). After training, we only keep the online encoder and use the output of the representation network as representations.

We use the PyTorch implementation of BYOL [2]. We use the representation network architecture as shown in Table 2, where the representation dimension $D$ is a parameter to be set. Except for normalization and output dimensions, the representation network architecture of BYOL and the encoder architecture of DisVAEs are similar. As for predictor and projector, we use the pipeline Linear$\rightarrow$ BN $\rightarrow$ ReLU $\rightarrow$ Linear with 256 hidden neurons. We train the BYOLs for 105 epochs using the Adam optimizer with $\beta_1 = 0.9$, $\beta_2 = 0.999$, $\epsilon = 10^{-8}$, and learning rate ($lr$) as a variable parameter. For augmentation, we use the pipeline of Cao et al. (2022) (in PyTorch-style):

1. *RandomApply(transforms.ColorJitter($x_{jit}$, $x_{jit}$, $x_{jit}$, 0.2), p=0.8)*
2. *RandomGrayScale(p=$p_{gray}$)*
3. *RandomHorizontalFlip()*
4. *RandomApply(transforms.GaussianBlur((3,3), (1.0, 2.0)), p=0.2)*
5. *RandomResizeCrop(size=(64, 64), scale=($x_{crop}$, 1.0))*

The $x_{\text{jit}}$, $p_{\text{gray}}$, and $x_{\text{crop}}$ are parameters to be set. $x_{\text{jit}}$ controls how much to jitter brightness, contrast, and saturation. $p_{\text{gray}}$ controls the probability to convert the image to grayscale. $x_{\text{crop}}$ defines the lower bound for the random area of the crop.

We perform a parameter sweep on the cross product of intervals of parameters $D$, $norm$, $lr$, $x_{\text{jit}}$, $p_{\text{gray}}$, and $x_{\text{crop}}$. On *3DShapes*, we use the following parameter grid (in scikit-learn style):

```
[
  {'D': [32, 64, 128], 'lr': [3e-2, 3e-3], 'norm': [BatchNorm()],
  'x_jit': [0.6, 0.8], 'p_gray': [0.5, 0.7, 0.9], 'x_crop': [1.0]},
  {'D': [256], 'lr': [3e-4, 3e-5],
  'norm': [BatchNorm(), GroupNorm(num_groups=4)], 'x_jit': [0.4, 0.8],
  'p_gray': [0.3, 0.5, 0.7], 'x_crop': [1.0]}
]
```

On *Abstract dSprites*, we use the following parameter grid:

---
[2] https://github.com/lucidrains/byol-pytorch.git

```
[
  {'D': [32, 64, 128], 'lr': [3e-3, 3e-4], 'norm': [BatchNorm()],
   'x_jit': [0.6, 0.8], 'p_gray': [0.0, 0.1, 0.2], 'x_crop': [0.6]},
  {'D': [256], 'lr': [3e-4, 3e-5],
   'norm': [BatchNorm(), GroupNorm(num_groups=4)], 'x_jit': [0.4, 0.8],
   'p_gray': [0.0, 0.1, 0.2], 'x_crop': [0.6]}
]
```

For each parameter configuration, we run it with 3 random seeds. Finally, we trained 360 BYOLs in total.

## A.2 ABSTRACT REASONING METHODS

We include two abstract reasoning network architectures: WReN (Barrett et al., 2018; van Steenkiste et al., 2019) and Transformer (Vaswani et al., 2017; Hahne et al., 2019).

**WReN implementation.** WReN consists of two parts: graph MLP and edge MLP. Here we use the same notations as in Section 3.3. For the representations of a trial $\text{Stage1}(T_i)$, edge MLP takes a pair of representations in $\text{Stage1}(T_i)$ as input and embed them to edge embeddings. Then all edge embeddings of $\text{Stage1}(T_i)$ (in total $C_9^2$=36) are added up and input to the graph MLP. Finally, the graph MLP output a scalar score, predicting the correctness of the trial $T_i$.

We use the code (van Steenkiste et al., 2019) to implement WReN. And we use the same parameter searching spaces as them. All WReNs are trained in 10K steps with a batch size of 32. The learning rate for the Adam optimizer is sampled from the set $\{0.01, 0.001, 0.0001\}$ while $\beta_1 = 0.9$, $\beta_2 = 0.999$, and $\epsilon = 10^{-8}$. For the edge MLP in the WReN model, we uniformly sample its hidden units in 256 or 512, and we uniformly choose its number of hidden layers in 2, 3, or 4. Similarly, for the graph MLP in the WReN model, we uniformly sample its hidden units in 128 or 512, and we uniformly choose its number of hidden layers in 1 or 2 before the final linear layer to predict the final score. We also uniformly sample whether we apply no dropout, dropout of 0.25, dropout of 0.5, or dropout of 0.75 to units before this last layer.

**Transformer implementation.** We simplify the architecture of Hahne et al. (2019). Here we treat $\text{Stage1}(T_i)$ as a sequence. We first linear project all representations and prepend them with a learnable `[class]` token. We add them with learnable positional embeddings. Then they are input into a stack of Transformer blocks (Vaswani et al., 2017). Finally, an MLP predicts a scalar score from the class embedding of the final Transformer block.

We implement the Transformer architecture ourselves with utilities of the DisLib code base. All Transformers are trained for the same steps and same batch size as WReN, i.e., 10K steps with a batch size of 32. We use the Adam optimizer with weight decay and cosine learning rate scheduler. The learning rate for the Adam optimizer is uniformly selected from $\{5e-4, 6e-4, 7e-4\}$. The depth of Transformer blocks is uniformly set to be 2, 3, or 4. The dimensions of $q, k, v$ of the self-attention model are uniformly 32 or 64. The MLP head uses the same architecture and parameter space as the graph MLP in WReN. For other fixed parameters, please refer to our codes for details.

## A.3 REPRESENTATION METRICS

In the main text, we employ disentanglement and informativeness metrics to measure the properties of representations. Here we provide more details.

**Disentanglement metrics.** We use the setup and implementation of Locatello et al. (2019b). Here we briefly introduce the details of our considered metrics. Namely, *BetaVAE* score (Higgins et al., 2016), *FactorVAE* score (Kim & Mnih, 2018), *Mutual Information Gap* (Chen et al., 2018), *SAP* (Kumar et al., 2017), and *DCI Disentanglement* (Eastwood & Williams, 2018). The *BetaVAE* score and the *FactorVAE* score predict the intervened factor from representations to measure disentanglement. The *Mutual Information Gap* and *SAP* compute the gap in response for each factor between the two highest representation dimensions. The difference is that MIG measures mutual information while SAP measures classification accuracy. The *DCI Disentanglement* calculates the entropy of the relative importance of a latent dimension in predicting factors. We follow previous studies (Locatello et al., 2019b; van Steenkiste et al., 2019; Locatello et al., 2019a; Dittadi et al., 2020) to develop a Gradient Boosting Tree (GBT) for prediction during the *DCI Disentanglement* evaluation.

Though according to Eastwood & Williams (2018) any classifier could be used. As reported by Cao et al. (2022), the GBT takes hours to train from high-dimensional representations learned by BYOL. Thus we only report *DCI Disentanglement* score for DisVAEs.

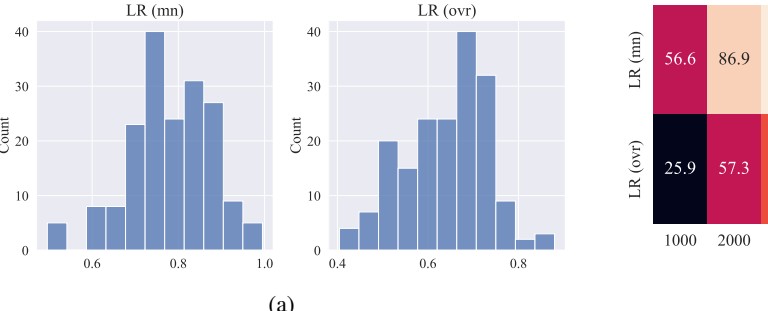

(a)                                                    (b)

Figure 9: (a) Prediction accuracy by "multinomial" LR, denoted as LR (mn), and "one v.s. rest" LR, denoted as LR (ovr) of DisVAEs on *3DShapes*. (b) LR (mn) and LR (ovr)'s correlations with the downstream performance of the DisVAEs-WReN pipeline.

**Informativeness metrics.** We use LR to measure the informativeness of representations. We train a *Logistic Regression* model to predict factor values from representations. We use 10000 samples to train LR. Unlike van Steenkiste et al. (2019), we use "multinomial" instead of "one v.s. rest" as the multi-class classification scheme. As shown in Figure 9a, for the same set of representations, "one v.s. Rest" LR has inferior prediction accuracy. Moreover, ranking by scores of these two LRs yields different results. In Figure ,9b we can observe different correlations of the "one v.s. Rest" LR. To better estimate informativeness, we use "multinomial" LR as the measurement.

### A.4  ABSTRACT VISUAL REASONING DATASETS

We use the two abstract visual reasoning datasets developed by van Steenkiste et al. (2019). i.e., Ravens' Progressive Matrices created from *3DShapes* (Burgess & Kim, 2018) and *Abstract dSprites* (Matthey et al., 2017; van Steenkiste et al., 2019).

We sketch the rules here by taking the RPM in Figure 1 as an example. The reasoning attributes are the ground truth factors of *3DShpaes*. i.e., floor hue, wall hue, object hue, scale, shape, and orientation. Each row in the $3 \times 3$ matrix has 1, 2, or 3 ground truth factors taking a fixed value. And the 3 rows have the same fixed ground truth factors, though they might take different values. From the context panels, one should discover the underlying logical relationship. Finally, one is asked to fill the missing panel by one of the candidates. For the RPM in Figure 1, from the contexts, we can infer that the fixed factors are: wall hue, shape, and orientation. Then for the third row, from the first 2 panels, we know that the values for the shared factors are: the wall hue is blue, the shape is cylinder, and the orientation is the azimuth that makes the wall corner appears in the righter part of the image. So we choose the candidate with these factor values as the solution, as shown in Figure 10a. Figure 10b shows a sample of RPMs with answers on *Abstract dSprites*.

## B  ABLATIONS ON GENERAL-PURPOSE REPRESENTATION LEARNING METHODS

In the main text, we use BYOL as a representative of general-purpose representation learning methods. For completeness, here we introduce another general-purpose method, SimSiam (Chen & He, 2021). We modify the code of BYOL [3] to train SimSiams on *3DShapes* with the following parameter grid:

```
[
  {'D': [512], 'lr': [3e-4, 3e-5],
  'norm': [BatchNorm()], 'x_jit': [0.4, 0.8],
```

---
[3] https://github.com/lucidrains/byol-pytorch.git

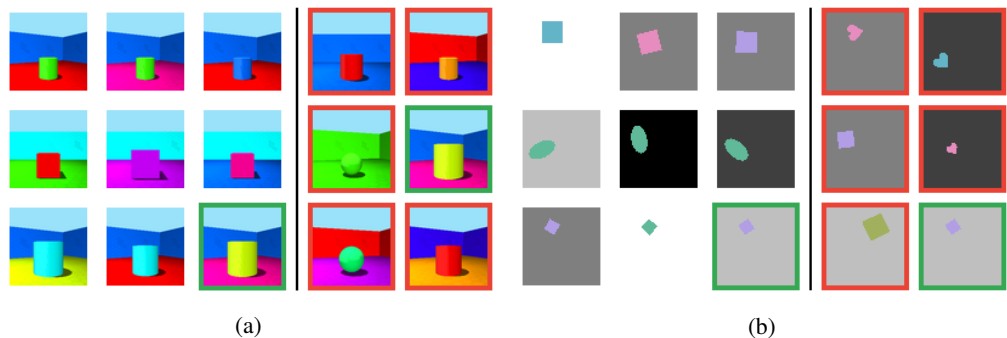

(a)                      (b)

Figure 10: RPM questions with solutions on (a) *3DShapes* and (b) *Abstract dSprites*.

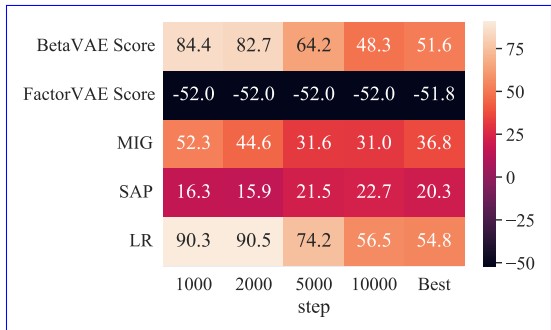

Figure 11: Correlations of SimSiam-WReN on *3DShapes*.

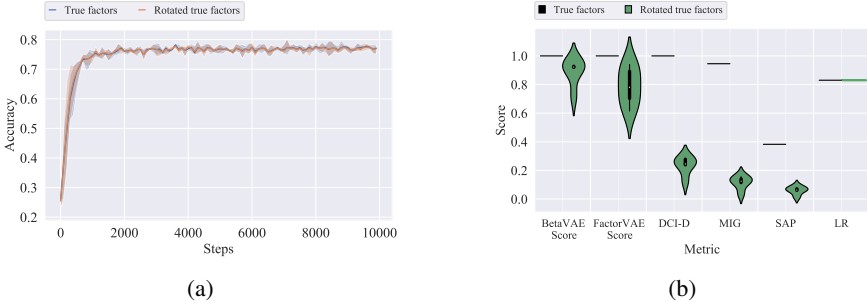

(a)                      (b)

Figure 12: (a) Average downstream accuracy throughout training from ground truth factors with and without rotation. (b) The violin plot of metric scores of original/rotated true factor values.

```
  'p_gray': [0.3, 0.5, 0.7], 'x_crop': [0.6, 1.0]}
]
```

For each configuration, we run with 3 seeds. So finally, we yield 72 SimSiams. Then we use the same WReNs for DisVAEs and BYOLs as Stage-2 models.

The results of SimSiam-WReN agree with our conclusions in the main text. As for the best performance, we have $\overline{\text{WReN}}$=85.1% and WReN$^\star$=94.1%, which is better than DisVAEs'. Figure 11 shows the correlations of downstream performance and representation properties. LR still correlates most for all considered steps.

## C  ADDITIONAL RESULTS

Table 3: The model type and the step achieving the performance in Table 1. We report in the form of model@step. For WReN and Trans., we report the mean steps and STDs.

| Dataset | Stage1 | WReN$^\star$ | $\overline{\text{WReN}}$ | Trans.$^\star$ | $\overline{\text{Trans.}}$ |
|---------|--------|--------------|--------------------------|----------------|----------------------------|
| *3DShapes* | DisVAEs | AnnealedVAE @9600 | $\beta$-VAE @8400(712) | DIP-VAE-I @9900 | $\beta$-TCVAE @9100(901) |
| *3DShapes* | BYOL | BYOL @10000 | BYOL @8900(849) | BYOL @8600 | BYOL @8860(937) |
| *Abstract dSprites* | DisVAEs | $\beta$-VAE @9900 | DIP-VAE-I @8920(898) | DIP-VAE-I @9400 | DIP-VAE-I @9380(172) |
| *Abstract dSprites* | BYOL | BYOL @8100 | BYOL @8320(1195) | BYOL @8800 | BYOL @8100(725) |

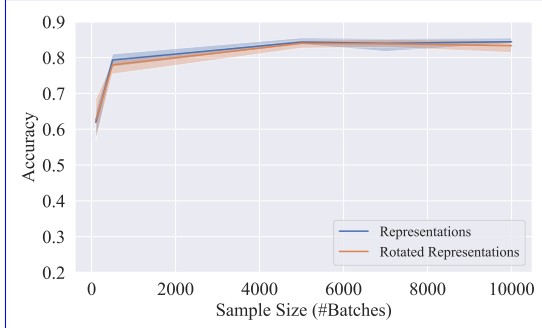

Figure 13: Accuracy v.s. #samples curves of the most disentangled DisVAEs before and after rotation. It is consistent with Figure 3.

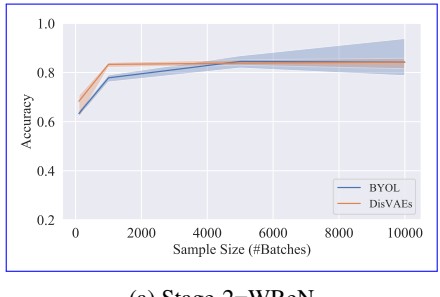

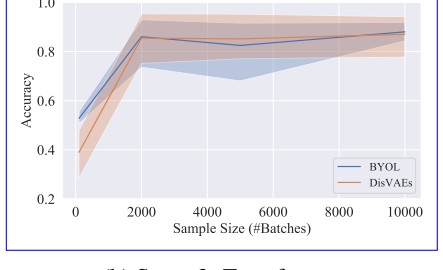

(a) Stage-2=WReN

(b) Stage-2=Transformer

Figure 14: Accuracy v.s. #samples curves of the Stage-1 models with the best $\overline{\text{WReN}}$ or $\overline{\text{Trans.}}$. It is consistent with Figure 4

## C.1 ADDITIONAL RESULTS OF FINAL PERFORMANCE

In Table 1 we report the best final performance of DisVAEs and BYOLs. Here we provide more details on which type of DisVAEs at which steps achieve the reported performance in Table 1. We can observe that the best DisVAEs vary with different datasets and Stage-2 models. As for the best steps, except *3DShapes*-WReN, BYOL achieves the best performance earlier than DisVAEs.

## C.2 ACCURACY-#SAMPLES CURVES

We employ training curves (accuracy-step) in the main text to evaluate sample efficiency following van Steenkiste et al. (2019). For completeness, here we show accuracy-#samples curves.

We present the accuracy-#samples versions of Figure 3 and Figure 4, i.e., Figure 13 and Figure 14. We train the same models as in the main text until convergence with fixed training data sizes of 100, 1000, 5000, 7000, and 10000 batches. Then for each sample size, we plot the test performance at the

Table 4: Mean metric scores with STDs of different Stage-1 models.

| Dataset | Stage1 | BetaVAE | FactorVAE | MIG | SAP | LR |
|---|---|---|---|---|---|---|
| *3DShapes* | DisVAEs | 93.7(7.7) | 82.3(11.1) | 25.5(15.1) | 6.5(3.8) | 78.0(9.6) |
| *3DShapes* | BYOL | 99.9(0.3) | 96.1 (4.6) | 8.1(5.3) | 1.2(0.9) | 96.6(1.8) |
| *Abstract dSprites* | DisVAEs | 62.3(14.1) | 49.1(10.5) | 13.3(7.0) | 6.8(3.4) | 36.8(4.4) |
| *Abstract dSprites* | BYOL | 63.6(17.0) | 62.4(11.8) | 2.6(1.8) | 0.5(0.3) | 43.0(8.2) |

step with the highest validation accuracy. We can see the ranking of representations and evolving patterns of both types of curves agree well.

## C.3 ADDITIONAL RESULTS OF RANDOM ROTATION EXPERIMENTS

This section contains additional results of the random rotation experiments. Here we report the downstream performance of deliberately entangled (by random rotation) representations.

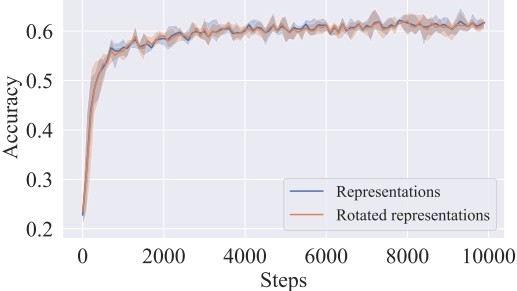

Figure 15: Test accuracy curves through training for representations before and after rotation on *Abstract dSprites*. Like for *3DShapes*, we select the DisVAEs with the highest *FactorVAE* score. The two curves are almost identical.

Figure 12 shows the same experiments as Figure 2 on *Abstract dSprites*. We can observe that the two curves in Figure 12a are almost identical. And in Figure 12b, we can observe that disentanglement metric scores drop drastically while LR remains the same. We notice that LR is not 100%. This is because some factors of *Abstract dSprites* have too many support values. e.g., the x and y positions both have 32 possible values. However, our conclusion in the main text still holds as we observe that LR is invariant to random rotation. On *Abstract dSprites*, we randomly rotate the most disentangled representations from DisVAEs (measured by *FactorVAE* score). In Figure 15, we can see that rotation has little impact on the training trajectories. So our conclusion is similar across datasets.

## C.4 ADDITIONAL RESULTS OF CORRELATIONS

In this part, we report additional results related to the correlations between representation metrics and downstream performance.

**Absolute values of metric scores and downstream accuracy.** We show the histograms as a sanity check of the distribution of metric scores and downstream accuracy. Figure 16 presents the score distributions of each metric. We report the mean metric scores with STDs to depict the overall properties for Stage-1 models in Table 4. Figure 17 and Figure 18 display the distributions of downstream performance.

**Rank correlations.** This part contains additional results of rank correlations. On *3DShapes*, Figure 19 displays rank correlations between adjusted metrics and downstream accuracy, Figure 20 shows the overall correlation between metrics. On *Abstract dSprites*, Figure 21 shows correlations between metrics and downstream performance. Then Figure 22 presents correlations between ad-

justed metrics and downstream performance. Finally, Figure 23 displays the overall correlations between metrics.

**Plots of (metric score, downstream accuracy) pairs.** Figures 24, 25, 26, 27, 28, 29, 30, and 31 provide an in-depth view of the correlations, where we plot (metrics, downstream accuracy) pairs.

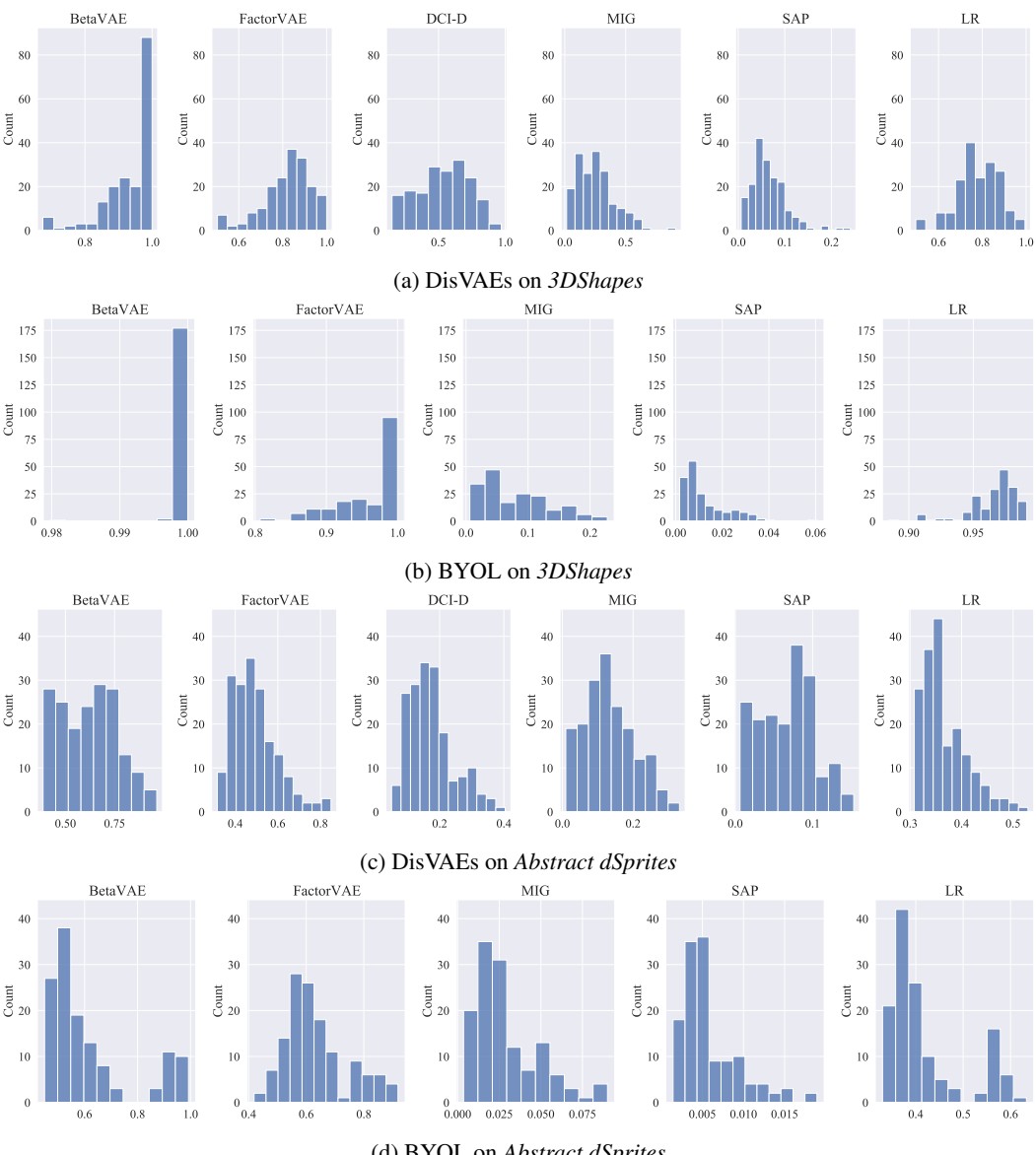

(a) DisVAEs on *3DShapes*

(b) BYOL on *3DShapes*

(c) DisVAEs on *Abstract dSprites*

(d) BYOL on *Abstract dSprites*

Figure 16: Histograms of metric scores of DisVAEs and BYOL on *3dShapes* and *Abstract dSprites*.

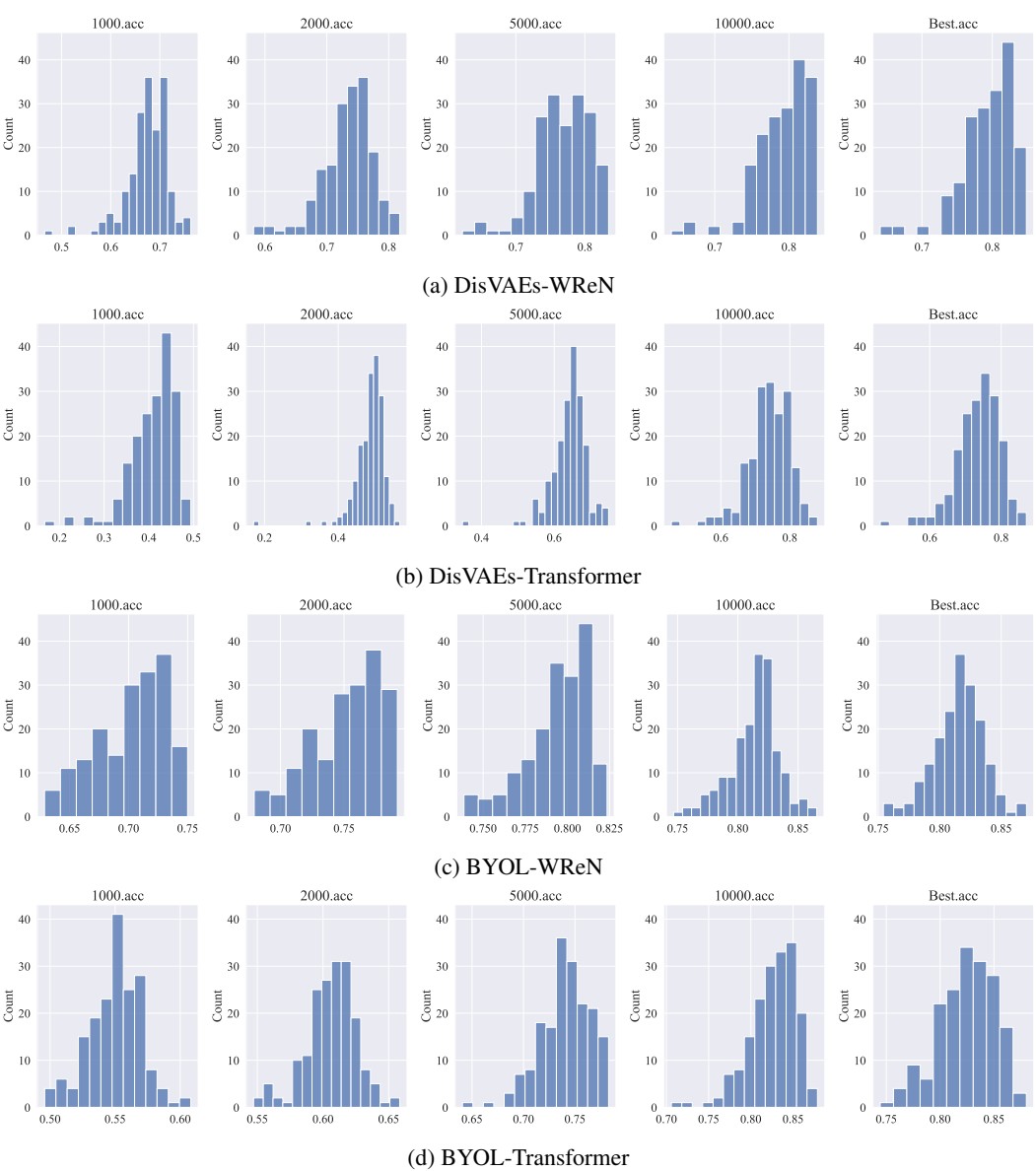

Figure 17: Histograms of downstream accuracy for multiple steps on *3DShapes*.

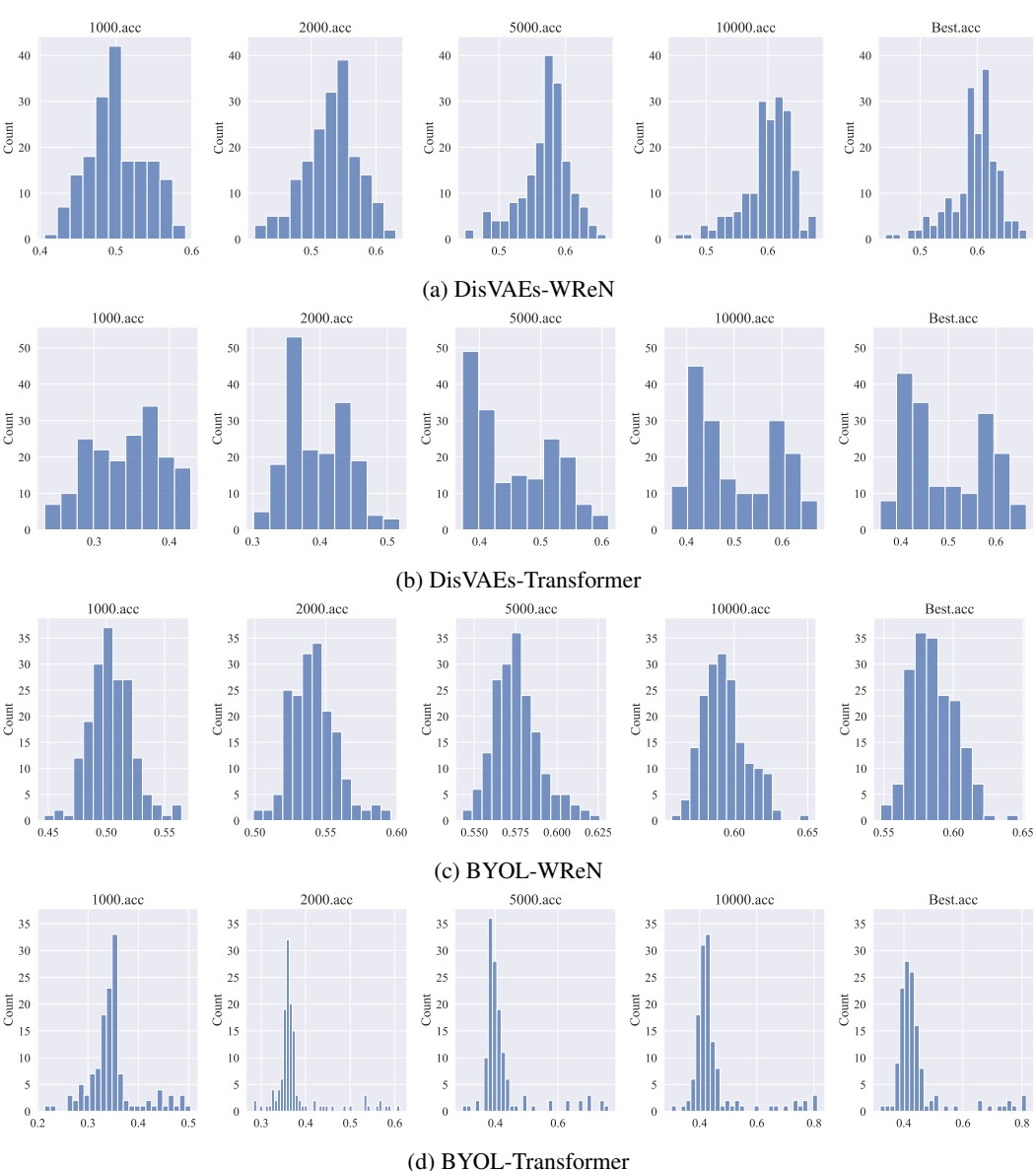

Figure 18: Histograms of downstream accuracy for multiple steps on *Abstract dSprites*.

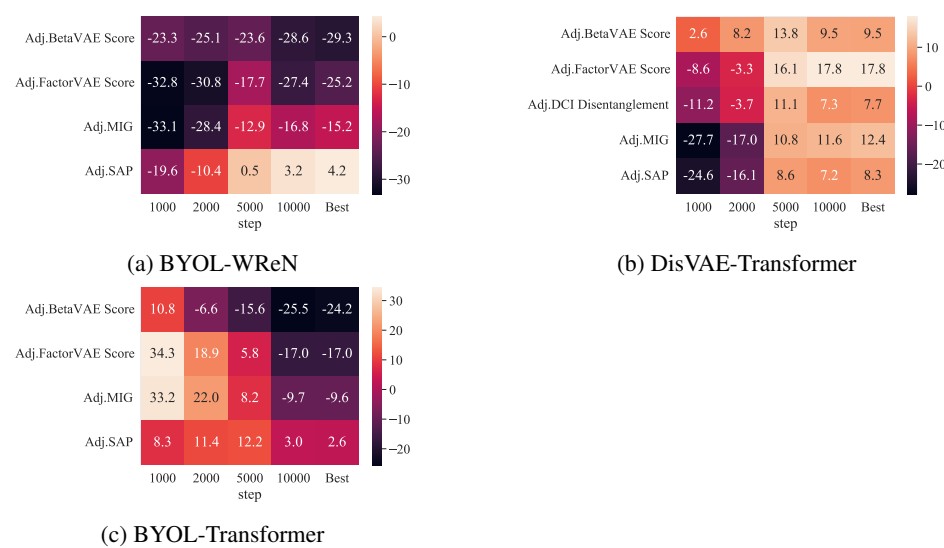

(a) BYOL-WReN

(b) DisVAE-Transformer

(c) BYOL-Transformer

Figure 19: Rank correlation between $\overline{\text{WReN}}$ or $\overline{\text{Trans.}}$ and adjusted metric scores on *3DShapes*.

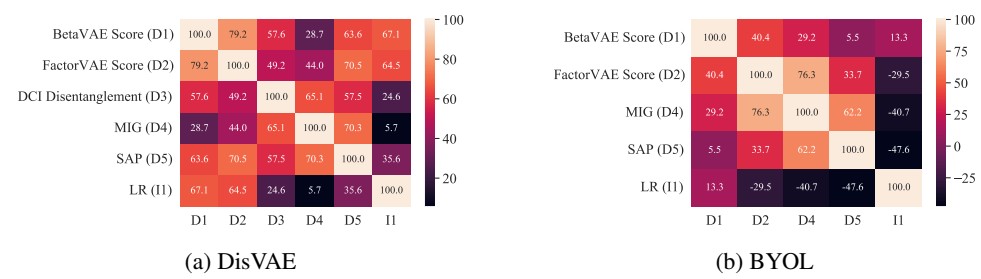

(a) DisVAE

(b) BYOL

Figure 20: Overall correlation between metric scores on *3DShapes*.

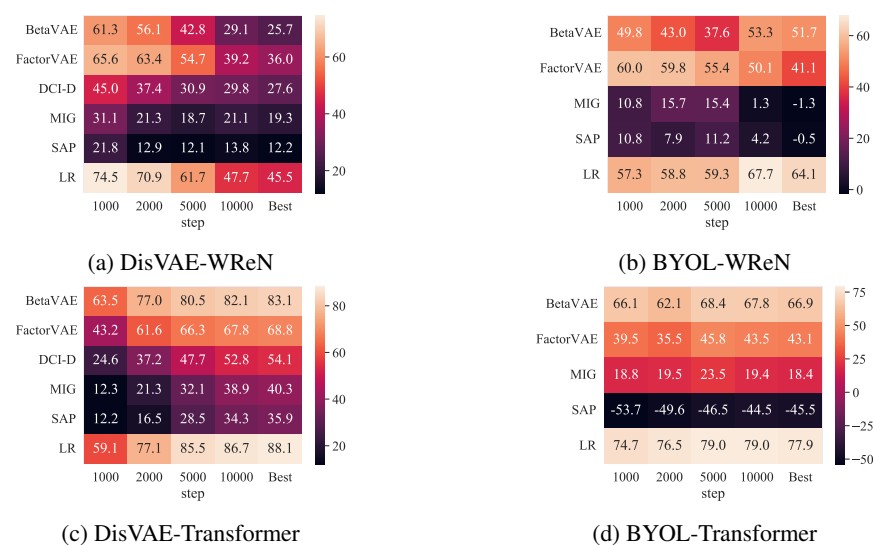

(a) DisVAE-WReN

(b) BYOL-WReN

(c) DisVAE-Transformer

(d) BYOL-Transformer

Figure 21: Rank correlation between $\overline{\text{WReN}}$ or $\overline{\text{Trans.}}$ and metric scores on *Abstract dSprites*.

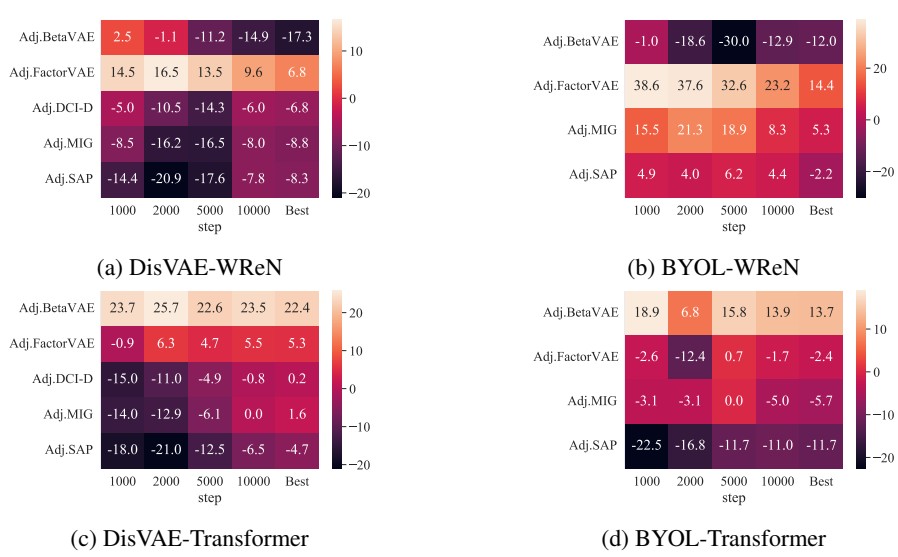

(a) DisVAE-WReN

(b) BYOL-WReN

(c) DisVAE-Transformer

(d) BYOL-Transformer

Figure 22: Rank correlation between $\overline{\text{WReN}}$ or $\overline{\text{Trans.}}$ and adjusted metric scores on *Abstract dSprites*.

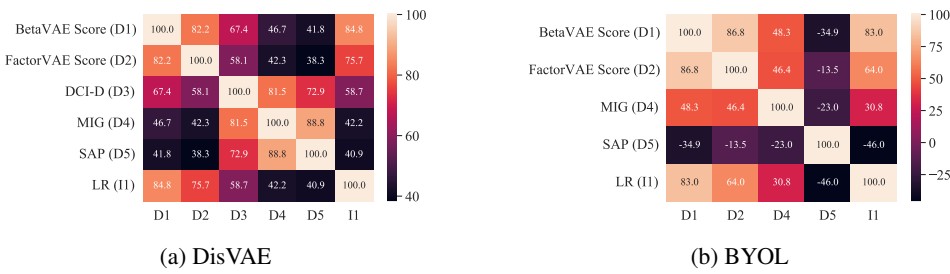

(a) DisVAE

(b) BYOL

Figure 23: Overall correlation between metric scores on *Abstract dSprites*.

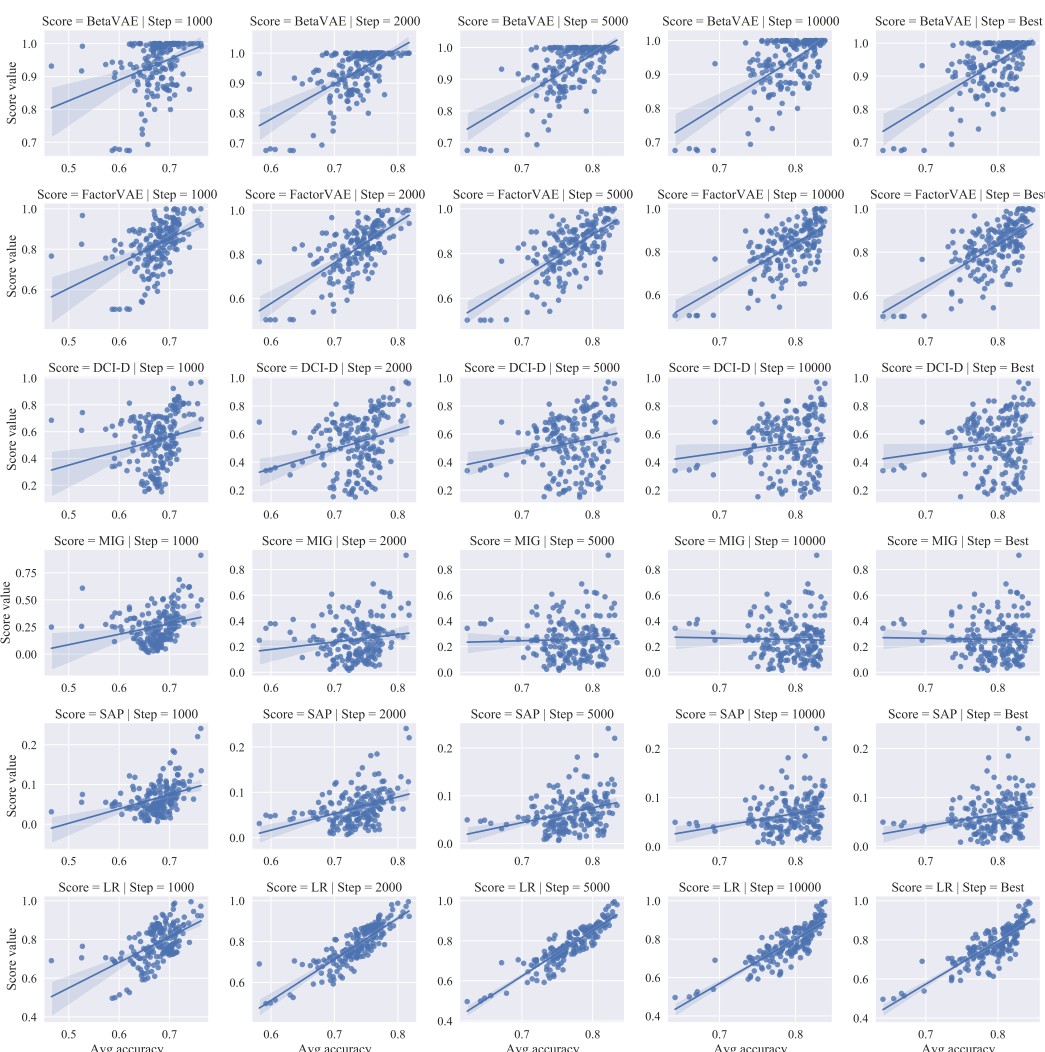

Figure 24: DisVAEs' metric scores v.s. $\overline{\text{WReN}}$ on *3DShapes*.

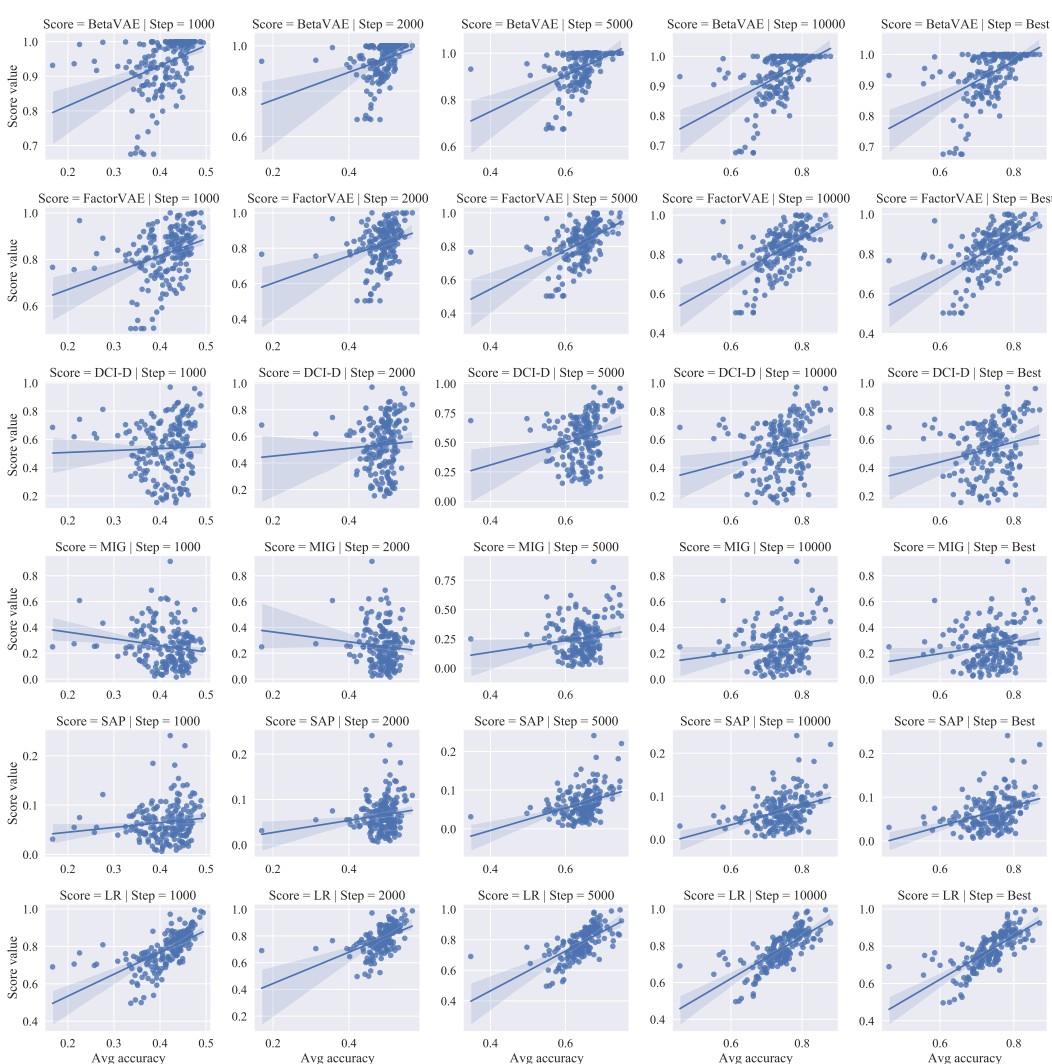

Figure 25: DisVAEs' metric scores v.s. $\overline{\text{Trans.}}$ on *3DShapes*.

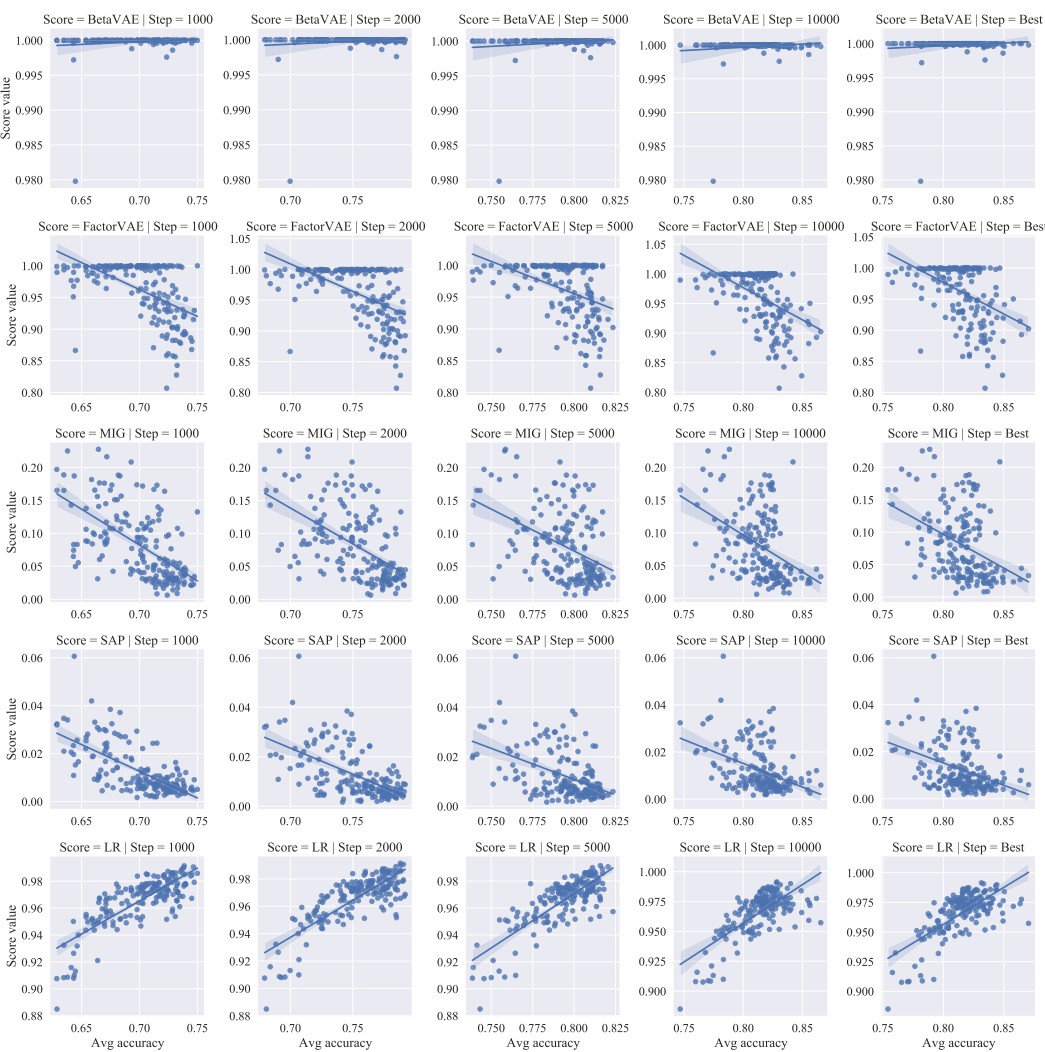

Figure 26: BYOLs' metric scores v.s. $\overline{\text{WReN}}$ on *3DShapes*.

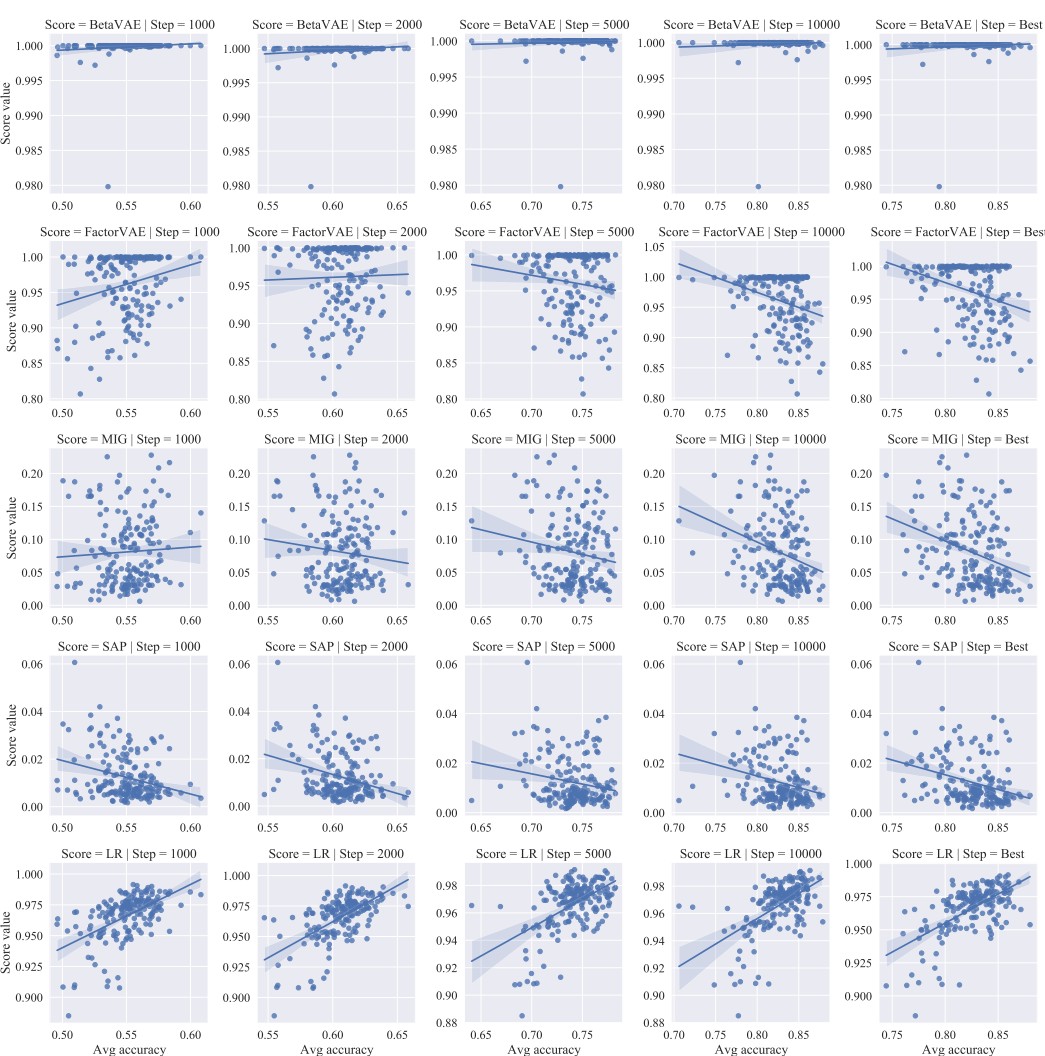

Figure 27: BYOLs' metric scores v.s. T̄r̄ā̄n̄s̄. on *3DShapes*.

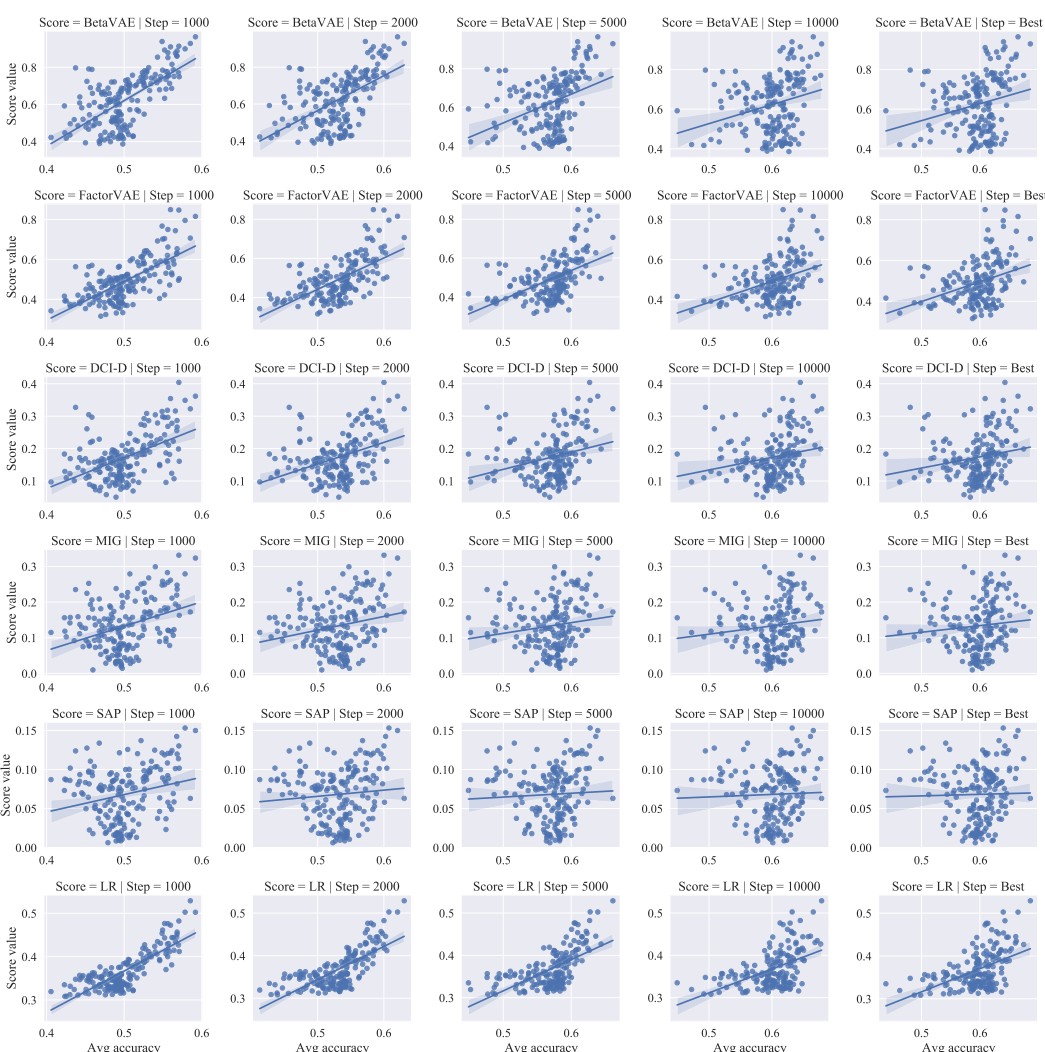

Figure 28: DisVAEs' metric scores v.s. $\overline{\text{WReN}}$ on *Abstract dSprites*.

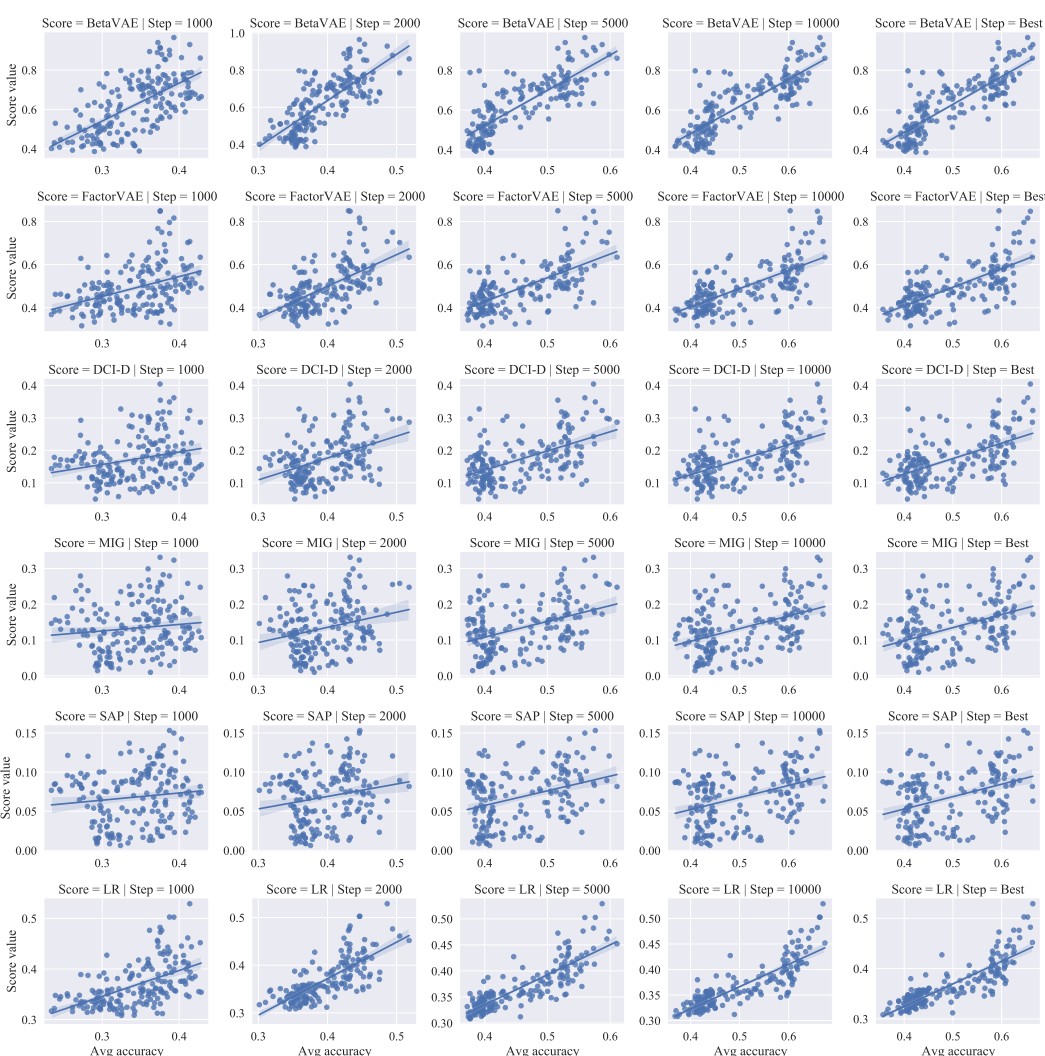

Figure 29: DisVAEs' metric scores v.s. $\overline{\text{Trans.}}$ on *Abstract dSprites*.

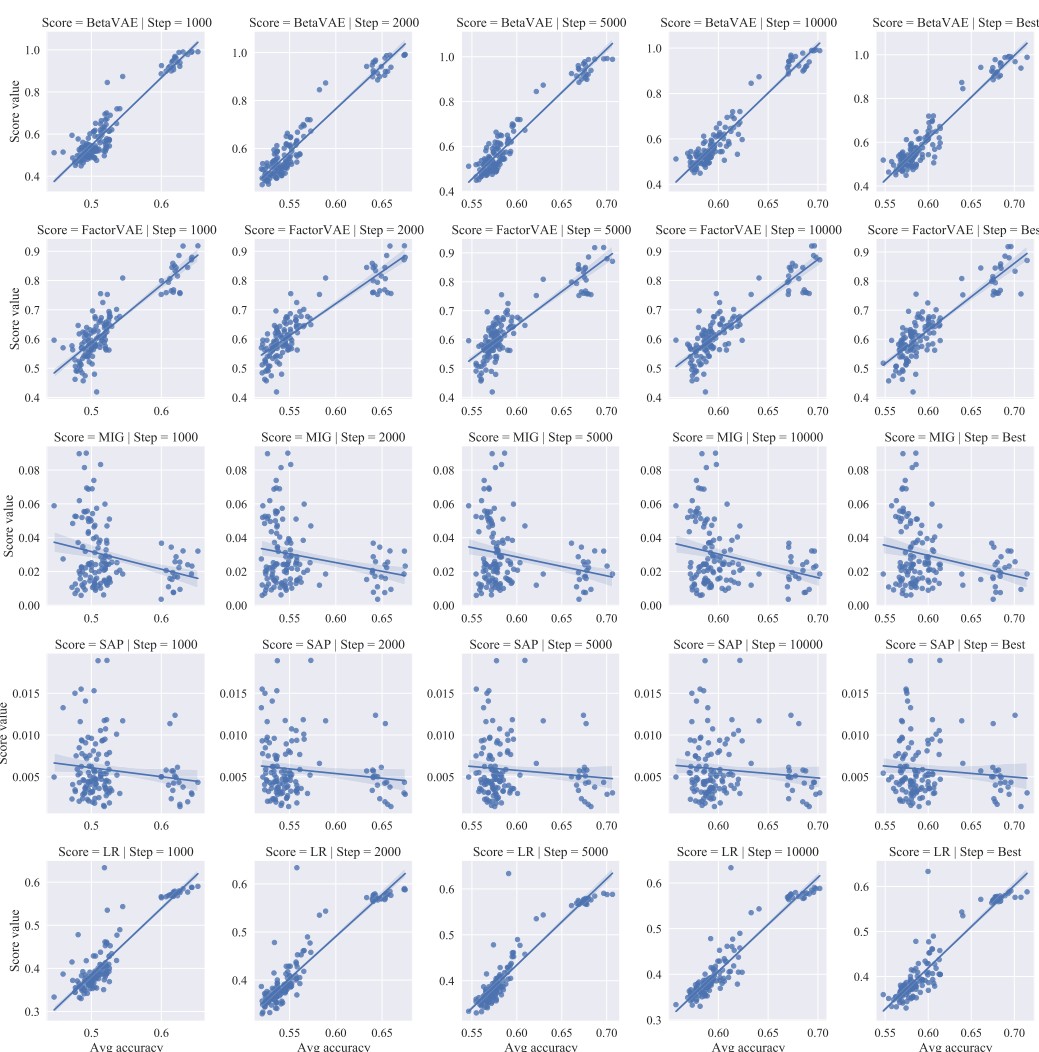

Figure 30: BYOLs' metric scores v.s. $\overline{\text{WReN}}$ on *Abstract dSprites*.

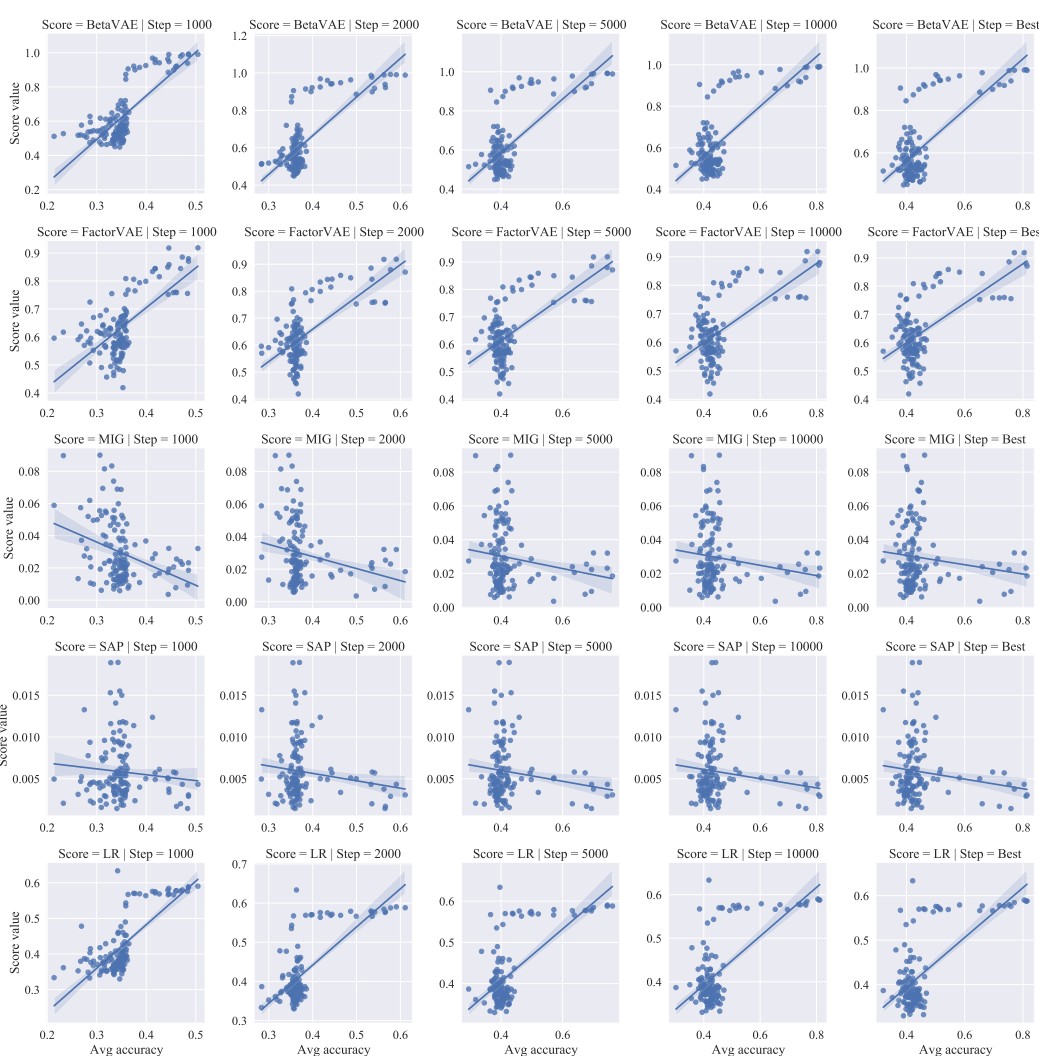

Figure 31: BYOLs' metric scores v.s. $\overline{\text{Trans.}}$ on *Abstract dSprites*.

