# OpenReview forum: "On the Necessity of Disentangled Representations for Downstream Tasks"
_ICLR.cc/2023/Conference — Submitted to ICLR 2023_

### Official Review · Reviewer_ZFjG · 2022-10-24

**Confidence:** 4
**Clarity, Quality, Novelty And Reproducibility:** The paper is well-written and clear
**Correctness:** 3
**Technical Novelty And Significance:** 2
**Empirical Novelty And Significance:** 3
**Recommendation:** 6

**Strength And Weaknesses:**

Strong points:

-extensive experimentation and evaluation

-effective use of logistic regression as a measure of informativeness

-a comparison between the effects of disentangled representation vs. deliberate entanglement of disentangled representations on downstream task

Weak points:

-lack of theoretical contribution

-there are far more types of disentangling VAEs against one general-purpose representation learning method (BYOL). Since downstream test performance is averaged over different VAEs, is this a fair comparison?

**Summary Of The Paper:**

This paper challenges a common belief that a disentangled representation is useful for downstream tasks. Following up Steenkiste et al., 2019 and Locatello et al. 2019b,  the authors focused on the informativeness of the representation and its correlation with the performance of downstream tasks.

**Summary Of The Review:**

Adding logistic regression as a measure of informativeness to previous measures of disentanglement, it is clear from experiments that this measure should be of value in evaluating learned representations in connection with downstream accuracy.

---

> ### Author Response · Authors · 2022-11-16
> **Response to Reviewer ZFjG**
>
> Dear reviewer,
>
> We are grateful for your reviews! Here is a point-by-point response to your comments and concerns.
>
> **Q1: Lack of theoretical contribution.**
>
> **A1:** Thanks for pointing this out.  Our rotation experiments are inspired by theoretical understanding. I.e., rotations will destroy disentanglement but not hurt informativeness. We start from this to challenge the necessity of disentanglement.
>
> Many theoretical frameworks of deep learning require extended empirical evidence. Take contrastive learning as an example. The impressive performance of empirical works like MoCo [1] and SimCLR [2] pave the way for the well-recognized alignment-uniformity theory [3].  The research on disentanglement's impacts on downstream tasks is mainly empirical so far (see Section 2).  We believe our findings are helpful. Since more observations are required to make reasonable assumptions and establish a theoretical framework.
>
> **Q2: There are far more types of disentangling VAEs against one general-purpose representation learning method (BYOL). Since downstream test performance is averaged over different VAEs, is this a fair comparison?**
>
> **A2:** It is fair. The downstream test performance is *maximum* over different DisVAEs. For the comparison of stage-1 models, the numbers of DisVAEs and BYOLs are the same (both are 360). They have the same amount of candidates. Moreover, their metric scores both exhibit reasonable spreads (see histograms in Section C.4).
>
> And according to Section A.1,  since DisVAEs have an extra hyper-parameter axis, i.e., model types, their choices are more flexible than BYOL's. Therefore we think when comparing the best performance (Table 1), DisVAEs are slightly more advantageous. Despite this, BYOL's performances are still comparable or better, which enhances our conclusion.
>
> For completeness, We have conducted experiments on another general-purpose learning model, SimSiam [4]. We have included the results of 72 SimSiams on 3DShapes with WReN as stage-2 models in Section B.  SimSiams achieve better performance than DisVAEs. As for the correlations, LR still correlates with the downstream accuracy most through all considered steps. Therefore, our conclusions still hold for SimSiam, i.e., disentanglement is not necessary.
>
> [1] He, K., Fan, H., Wu, Y., Xie, S., & Girshick, R. (2020). Momentum contrast for unsupervised visual representation learning. In *Proceedings of the* *IEEE**/CVF conference on* *computer vision and pattern recognition* (pp. 9729-9738).
>
> [2] Chen, T., Kornblith, S., Norouzi, M., & Hinton, G. (2020, November). A simple framework for contrastive learning of visual representations. In *International conference on* *machine learning* (pp. 1597-1607). PMLR.
>
> [3] Wang, T., & Isola, P. (2020, November). Understanding contrastive representation learning through alignment and uniformity on the hypersphere. In *International Conference on* *Machine Learning* (pp. 9929-9939). PMLR.
>
> [4] Dittadi, A., Träuble, F., Locatello, F., Wüthrich, M., Agrawal, V., Winther, O., ... & Schölkopf, B. (2021). On the Transfer of Disentangled Representations in Realistic Settings. In *International Conference on Learning Representations*.
>
> [5] Locatello, F., Abbati, G., Rainforth, T., Bauer, S., Schölkopf, B., & Bachem, O. (2019). On the fairness of disentangled representations. *Advances in Neural Information Processing Systems*, *32*.
>
> [6] Chen, X., & He, K. (2021). Exploring simple siamese representation learning. In *Proceedings of the* *IEEE**/CVF Conference on* *Computer Vision and Pattern Recognition* (pp. 15750-15758).

---

### Official Review · Reviewer_Pb5b · 2022-10-24

**Confidence:** 5
**Correctness:** 2
**Technical Novelty And Significance:** 2
**Empirical Novelty And Significance:** 3
**Recommendation:** 5

**Clarity, Quality, Novelty And Reproducibility:**

The claims discussed in the paper are relevant for the research community. The experiments seems well-designed, except for the points highlighted in the previous section. The writing style is generally clear and concise. The authors report all the details needed for reproducibility; however, they do not share the code of the experiments (but commit to sharing it after publication).

**Strength And Weaknesses:**

STRENGHTS

(+) The experimental setting, is for the most part, reasonable and well-designed.  The paper is well-written and easy to follow. The experiments try to tackle relevant questions in the representation learning community.

WEAKNESSES

The evidence presented is not enough to back up all claims of this paper. In particular, I have the following concerns:

(-) In the first contribution, the true disentangled representations are compared only with a rotated version of themselves. Even if the rotated representations are not disentangled according to the metrics, I think that they are still a much more similar to a disentangled representation than the typical representations learned by deep learning models. Therefore, comparing between the two is not enough to claim that disentanglement is not beneficial for downstream performance in general. A fairer comparison would be between true disentangled representations and entangled representations learned by a standard VAE, where \lambda_1 and \lambda_2  of Equation 3 are both set to 0.

(-) Table 1. Two concerns: first, for which DisVAEs model does the reported accuracy refer to? Is it an average between the models? Second, reporting the step with the highest validation accuracy for both DisVAE and BYOL can be misleading, since it hides information about *when* that accuracy is achieved. It might be the case where BYOL achieves an overall better accuracy, but much later than DisVAEs. In that case, using a DisVAE model for representation learning could still be valuable. It is generally fairer to fix a specific number of training steps, or to run an early stopping strategy. More generally, it can be interesting to have more information about the speed of convergence of these model. Furthermore, I would have expected to see the standard deviations of performance of the 5 different runs of WReNs and Transformers reported in Table 1.

(-) Figure 1 should be moved closer to section 4.2, for the sake of readability.The results of  Figures 5 and 6 show that informativeness has a strong correlation with downstream tasks. However, it seems that disentanglement somewhat implies informativeness, and that the representations trained with BYOLS exhibit high disentanglement scores on dSprites, at least in the case of the beta-VAE and FactorVAE scores. I am not sure that correlations alone are enough to make any conclusions about the usefulness of disentanglement. In order to conclude that informativeness is more beneficial than disentanglement on downstream tasks, I would like to see a comparison of the absolute values of disentanglement scores and informativeness scores for the two representations, showing that BYOL representations have higher informativeness, lower disentanglement, and higher downstream performance than DisVAE representations. Finally, it is still not very clear to me how the authors selected the final DisVAEs representation to be used in the figures.

On the minors side, Figure 1 should be moved closer to section 4.2, for the sake of readability.

**Summary Of The Paper:**

The paper performs a large scale empirical study to investigate whether disentangled representations provide a clear benefit for the final performance on downstream tasks. First, the ground-truth disentangled representation (normalized true factors) are compared to a rotated version of the same representations. The authors show that the two types of representations yields no significant difference in the final downstream performance. The second paper contribution compares the final performance of two models (a WReN and a Transformer) on an abstract reasoning tasks, using representations learned both via disentanglement-oriented learning methods (DisVAEs) and an entangled representation learning method (BYOL). They report a better downstream performance using representations based on BYOL. Finally, the authors show that the Informativeness is the metric that correlates the most with downstream performance on both DisVAE and BYOL representations, with disentanglement bringing only a small extra benefit.

**Summary Of The Review:**

This paper tries to tackle some relevant problems of the representation learning community. While the experiments are generally well-designed, it does not seem to me that the presented evidence is enough to support the major claims of this paper.  Some additional experiments are needed, and some concerns in the experimental settings need to be addressed, in order to make this manuscript suitable for publication.

Minor points:
 - The term “FactorVAE” is often spelled incorrectly on page 8.
 - The second point of contributions in the introduction start with the sentence “We show that what information...” that is a bit hard to read. I would recommend rephrasing that sentence.

---

> ### Author Response · Authors · 2022-11-16
> **Response to Reviewer Pb5b: (1) Comparison between representations and (2) concerns about Table 1**
>
> Dear reviewer,
>
> We are grateful for your reviews and suggestions. We will respond to your questions by points.
>
> **Q1:**  **A fairer comparison would be between true disentangled representations and entangled representations learned by a standard VAE.**
>
> **A1:** Thanks for your suggestion. This comparison is unfair since *true disentangled representations* are not learned by neural networks. Instead, it is normalized ground truth factor values, which are taken from the labels of the dataset.
>
> Moreover, disentanglement is not the primary variable in this comparison. There is a large gap between their informativeness. The informativeness of true disentangled representations is 100%.  But in our cases, the average informativeness of DisVAEs is 78.0% on 3DShapes and 36.8% on abstract dSprites.
>
> In our study, we perform the comparison between typical representations learned by deep neural networks (not entangled by rotation) by comparing BYOLs and DisVAEs. Empirically, BYOL learns less disentangled representations (see histograms in Section C.3, bellowing response to **Q3**, and [2]) but reaches comparable or better performance.
>
> **Q2:** **Concerns about Table 1.**
>
> **Q2.1: Which DisVAEs model does the reported accuracy refer to?**
>
> **A2.1:** The results in Table1 are the maximum over *all* DisVAEs. Taking DisVAEs-WReN on 3DShapes as an example, each of the 180 DisVAEs is scored with a mean performance and a max performance (over 5 WReNs learn from its representation), i.e., $\overline{\operatorname{WReN}}$, and $\operatorname{WReN}^{\star}$. Then in Table1, we report the highest scores among these 180 DisVAEs.
>
> **Q2.2: Reporting the step with the highest validation accuracy for both DisVAE and BYOL can be misleading.**
>
> **A2.2:** Thanks for your suggestions. Please refer to Figure 4 for the downstream learning process of DisVAEs and BYOL. We select the models with the highest $\overline{\operatorname{WReN}}$ or $\overline{\operatorname{Trans.}}$ (the models in the corresponding columns in Table1) to plot the test accuracy v.s. training steps curves. Thus the steps achieving the best training accuracy and the convergence speed can be read from the curves. We can see that the curves of DisVAEs and BYOL are close to each other. Therefore there is no significant gap between their sample efficiency and final performance.
>
> In the following table, we report the DisVAE classes that reach the best performance, together with the steps achieving the best validation accuracy in the `model@step` form. For $\overline{\operatorname{WReN}}$ and $\overline{\operatorname{Trans.}}$, we report the mean steps and their STDs in the `model@mean_step(STD)` form. As for the best steps, except 3DShapes-WReN, we can observe BYOL achieves the best performance earlier than DisVAEs. Moreover, the best DisVAE family varies with datasets and downstream methods. We have included this table in  Section C.1.
>
> | Dataset             | Stage1  | $\operatorname{WReN}^\star$ | $\overline{\operatorname{WReN}}$ | $\operatorname{Trans.}^\star$ | $\overline{\operatorname{Trans.}}$ |
> | ------------------- | ------- | --------------------------- | -------------------------------- | ----------------------------- | ---------------------------------- |
> | *3DShapes*          | DisVAEs | AnnealedVAE@9600            | $\beta$-VAE@8400(712)            | DIP-VAE-I@9900                | $\beta$-TCVAE@9100(901)            |
> | *3DShapes*          | BYOL    | BYOL@10000                  | BYOL@8900(849)                   | BYOL@8600                     | BYOL@8860(937)                     |
> | *Abstract dSprites* | DisVAEs | $\beta$-VAE@9900            | DIP-VAE-I@8920(898)              | DIP-VAE-I@9400                | DIP-VAE-I@9380(172)                |
> | *Abstract dSprites* | BYOL    | BYOL@8100                   | BYOL@8320(1195)                  | BYOL@8800                     | BYOL@8100(725)                     |
>
> **Q2.3: STDs in Table1.**
>
> **A2.3:** Thanks for your suggestion. We have included standard deviations in Table 1. We find no apparent link between the type of stage-1 models and STDs.

---

> > ### Author Response · Authors · 2022-11-16
> > **Response to  Reviewer Pb5b: (3) Concerns about the figures and (4) Minors**
> >
> > **Q3: Concerns about the figures.**
> >
> > **Q3.1: Moving Figure1 closer to Section 4.2.**
> >
> > **A3.1:** Thanks for your suggestion. We have moved the previous Figure 1 (now Figure 2) to the same page as Section 4.2.
> >
> > **Q3.2: Comparing the absolute values of metric scores and downstream performance.**
> >
> > **A3.2:** Please refer to the histograms in Section C.4 for the absolute values of metric scores (Figure 16 and Figure 17). Here we also report their mean values and STDs in the following table.
> >
> > | Dataset             | Stage1  | BetaVAE    | FactorVAE  | MIG        | SAP      | LR        |
> > | ------------------- | ------- | ---------- | ---------- | ---------- | -------- | --------- |
> > | *3DShapes*          | DisVAEs | 93.7(7.7)  | 82.3(11.1) | 25.5(15.1) | 6.5(3.8) | 78.0(9.6) |
> > | *3DShapes*          | BYOL    | 99.9(0.3)  | 96.1 (4.6) | 8.1(5.3)   | 1.2(0.9) | 96.6(1.8) |
> > | *Abstract dSprites* | DisVAEs | 62.3(14.1) | 49.1(10.5) | 13.3(7.0)  | 6.8(3.4) | 36.8(4.4) |
> > | *Abstract dSprites* | BYOL    | 63.6(17.0) | 62.4(11.8) | 2.6(1.8)   | 0.5(0.3) | 43.0(8.2) |
> >
> >
> > It's true that BYOLs achieve higher disentanglement scores in terms of BetaVAE score and FactorVAE score. However, the different dimensionality of BYOL's and DisVAEs' representations (~$10^2$ v.s. $10$) lead to different scales of these two metrics. [2] conducted large-scale experiments showing BetaVAE score and FactorVAE score prefer higher dimensional representations. It is because these two metrics measure whether intervening in a factor results in a significant change in only one dimension. As BYOLs require a high-dimensional representation to train[3], there are more chances for BYOLs to have such a 'lucky' dimension when compared with DisVAEs.    Moreover, according to [2]'s empirical results, BYOL's representations are not well-disentangled.
> >
> > With similar scales of metric scores, intra-stage-1 comparisons are valid due to the rationale of these metrics and empirical evidence in [4]. Therefore we think rank correlations should be more appropriate to measure the usefulness of disentanglement. Notably, this is adopted by previous works [4-6].
> >
> > **Q3.3: How the authors selected the final DisVAEs representation to be used in the figures.**
> >
> > **A3.3:** For figures of rotation experiments (Figure 3, Figure 13, Figure 15), we use the DisVAE with the highest FactorVAE scores. For training curves of different stage-1 models (Figure 4, Figure 14), we select the model with the highest $\overline{\operatorname{WReN}}$ or $\overline{\operatorname{Trans.}}$. For results related to correlations (histograms, reg-plots, correlations), we use *all* models we trained.
> >
> > **Minor issues:**
> >
> > Thanks so much for pointing out the problems! We have corrected the typos on page 8 and rephrased our contribution (2).
> >
> > [1] M. Rolínek, D. Zietlow and G. Martius, "Variational Autoencoders Pursue PCA Directions (by Accident)," 2019 IEEE/CVF Conference on Computer Vision and Pattern Recognition (CVPR), 2019, pp. 12398-12407, doi: 10.1109/CVPR.2019.01269.
> >
> > [2] Cao, J., Nai, R., Yang, Q., Huang, J., & Gao, Y. (2022). An Empirical Study on Disentanglement of Negative-free Contrastive Learning. *arXiv preprint arXiv:2206.04756*.
> >
> > [3] Grill, J. B., Strub, F., Altché, F., Tallec, C., Richemond, P., Buchatskaya, E., ... & Valko, M. (2020). Bootstrap your own latent-a new approach to self-supervised learning. *Advances in neural information processing systems*, *33*, 21271-21284.
> >
> > [4] Van Steenkiste, S., Locatello, F., Schmidhuber, J., & Bachem, O. (2019). Are disentangled representations helpful for abstract visual reasoning?. *Advances in Neural Information Processing Systems*, 32.
> >
> > [5] Dittadi, A., Träuble, F., Locatello, F., Wüthrich, M., Agrawal, V., Winther, O., ... & Schölkopf, B. (2021). On the Transfer of Disentangled Representations in Realistic Settings. In *International Conference on Learning Representations*.
> >
> > [6] Locatello, F., Abbati, G., Rainforth, T., Bauer, S., Schölkopf, B., & Bachem, O. (2019). On the fairness of disentangled representations. *Advances in Neural Information Processing Systems*, *32*.

---

> > > ### Comment · Reviewer_Pb5b · 2022-11-18
> > > **Rebuttal discussion**
> > >
> > > Many thanks for the exaustive elaboration on my concerns: several things are now clearer an allow to focus on what are the remaining keypoints of my assessment.
> > >
> > > First, I am missing a convincing response my concern about  rotated representations being inadequate to make a representation entangled: e.g. nonlinear ICA might lead to representations that are equivalent up-to rotation. Overall I am  no convinced that results in Section 4.2 are convincing enough to demonstrate that disentanglement has no effect on downstream performance. Section 4.3 is a bit puzzling because instead of comparing disentanglement and informativeness on the same setting as in 4.2, a different set of experiments is run showing that one correlates with downstream tasks and the other less. This is a bit confusing and perhaps not really neat (unless I am missing some fundamental reason to have a totally different setting than the one that "demonstrates" that disentanglement is not helpful).
> > >
> > > In A3.2 the result hint at the fact that disentanglement metric are sensitive to dimensionality, which kinds of contrast with the paper idea. In betaVAE and Factor VAE are dimensionality sensitive, then dimensionality should be fixed to allow fair comparison.

---

> > > > ### Author Response · Authors · 2022-11-20
> > > > **Response to Reviewer Pb5b: concerns about (1) rotation and (2) disentanglement metrics**
> > > >
> > > > Dear Reviewer,
> > > >
> > > > Thanks for your questions. We would like to respond to you by points.
> > > >
> > > > **Q1: Rotated representations are inadequate to make a representation entangled**.
> > > >
> > > > **A1:** It is a well-known corollary of disentanglement's definition that rotation will destroy disentanglement [1, 2]. According to the definition, a representation dimension should only encode information from one factor. However, after rotation, a dimension will encode a combination of factors, i.e., the axis alignment is violated.
> > > >
> > > > ICA is related to representation disentanglement, but they pursue different goals. ICA requires *identifiability*, i.e., true factors can be recovered from representations (up to linear or non-linear transform). Therefore it is invariant to rotations. In contrast, representation disentanglement emphasizes the encoding pattern, i.e., the dimension-wise correspondence.
> > > >
> > > > In summary, rotation entangles representations by definition. Thus we use it as a counter-example to show that entangling representations does not hurt downstream performance.
> > > >
> > > > *We would like to first discuss issues of disentanglement metric scores to better address your concerns about Section 4.3.*
> > > >
> > > > **Q2: In betaVAE and Factor VAE are dimensionality sensitive, then dimensionality** **should be fixed to allow fair comparison.**
> > > >
> > > > **A2:** Thanks for pointing this out. We only calculate rank correlations in the same setting (representation learning methods, downstream model structures, and dataset). Thus the comparisons between those disentanglement scores are always among the same representation dimensionality. That is also the reason why we don't compare absolute scores but compare correlations instead.
> > > >
> > > > We note that those methods require specific numbers of dimensionality to work well. Thus we don't really have the option to compare with the same number of dimensions for each method. For example, BYOL works better with higher representation dimensionality [6, 7], while VAE methods do better with lower dimensions [6].

---

> > > > > ### Author Response · Authors · 2022-11-20
> > > > > **Response to Reviewer Pb5b: (3) concerns about Section 4.3**
> > > > >
> > > > > **Q3: Section 4.3 is a bit puzzling because instead of comparing disentanglement and informativeness** **on the same setting as in 4.2,** **a different set of experiments is run showing that** **one correlates with downstream tasks and the other less.**
> > > > >
> > > > > **A3:** Comparing absolute values like in Section 4.2 can be inadequate to demonstrate which of them is more significant. Because metric scores have different scales and offsets for DisVAEs and BYOLs. Only showing BYOL outperforms DisVAEs with higher informativeness scores and lower disentanglement scores can be inaccurate.  DisVAE with higher disentanglement scores (like BetaVAE score and FactorVAE score) than BYOL does not necessarily indicate they are more disentangled.
> > > > >
> > > > > The table in the previous **A3.2** has shown the comparison between absolute values of disentanglement and informativeness.  Except for BetaVAE score and FactorVAE score, the results of other metrics agree with expectations. I.e., BYOLs have higher informativeness scores and lower disentanglement scores and perform better than or comparable with DisVAEs.
> > > > >
> > > > > We think the rank correlation is a more appropriate measurement to compare the effects of disentanglement and informativeness. Sections 4.2 and 4.3 focus on different questions.  Section 4.2 aim to challenge the necessity of disentanglement. We construct counter-examples that entangled representations (re-entangled by rotation or learned by general-purpose models) can still perform comparable or better. While in Section 4.3, we analyze which property matters downstream performance. We aim to assess the significance of the relation between downstream accuracy and these two properties. Therefore we opt to use rank correlations to tell "whether more disentangled/informative representations lead to better downstream performance". Leveraging the large-scale empirical results, rank correlation provides an overall measurement of significance rather than focusing on individuals.
> > > > >
> > > > > Moreover, analyzing rank correlations is widely adopted by previous works about disentanglement's effects on downstream tasks [3, 4, 5].
> > > > >
> > > > > [1] Bengio, Y., Courville, A., & Vincent, P. (2013). Representation learning: A review and new perspectives. *IEEE* *Transactions on Pattern Analysis and Machine Intelligence*, 35(8), 1798-1828.
> > > > >
> > > > > [2] Locatello, F., Bauer, S., Lucic, M., Rätsch, G., Gelly, S., Schölkopf, B., & Bachem, O. (2020). A Sober Look at the Unsupervised Learning of Disentangled Representations and their Evaluation. *Journal of Machine Learning Research*, 21, 1-62.
> > > > >
> > > > > [3] Van Steenkiste, S., Locatello, F., Schmidhuber, J., & Bachem, O. (2019). Are disentangled representations helpful for abstract visual reasoning?. *Advances in Neural Information Processing Systems*, 32.
> > > > >
> > > > > [4] Dittadi, A., Träuble, F., Locatello, F., Wüthrich, M., Agrawal, V., Winther, O., ... & Schölkopf, B. (2021). On the Transfer of Disentangled Representations in Realistic Settings. In *International Conference on Learning Representations*.
> > > > >
> > > > > [5] Locatello, F., Abbati, G., Rainforth, T., Bauer, S., Schölkopf, B., & Bachem, O. (2019). On the fairness of disentangled representations. *Advances in Neural Information Processing Systems*, *32*.
> > > > >
> > > > > [6] Cao, J., Nai, R., Yang, Q., Huang, J., & Gao, Y. (2022). An Empirical Study on Disentanglement of Negative-free Contrastive Learning. *arXiv preprint arXiv:2206.04756*.
> > > > >
> > > > > [7] Grill, J. B., Strub, F., Altché, F., Tallec, C., Richemond, P., Buchatskaya, E., ... & Valko, M. (2020). Bootstrap your own latent-a new approach to self-supervised learning. *Advances in neural information processing systems*, *33*, 21271-21284.

---

> > > > > > ### Author Response · Authors · 2022-11-25
> > > > > > **Do our responses address your concerns?**
> > > > > >
> > > > > > Dear Reviewer Pb5b,
> > > > > >
> > > > > > We would like to ask if our responses have adequately addressed your concerns and answered your questions. If not, we are happy for any further discussions.
> > > > > >
> > > > > > Regards,
> > > > > >
> > > > > > The authors.

---

> > > > > > > ### Comment · Reviewer_Pb5b · 2022-11-28
> > > > > > > **Helpful clarifications**
> > > > > > >
> > > > > > > Dear Authors,
> > > > > > > many thanks for the fruitful discussion. The responses you provided have made the extent of your contribution much clear to me. I will take all your comments into consideration in the internal discussion with the reviewers.
> > > > > > > Regards.

---

### Official Review · Reviewer_hRSM · 2022-10-25

**Confidence:** 4
**Clarity, Quality, Novelty And Reproducibility:** I don’t have specific concerns on cla…
**Correctness:** 4
**Technical Novelty And Significance:** 3
**Empirical Novelty And Significance:** 3
**Recommendation:** 6

**Strength And Weaknesses:**

Strength:

--The authors challenge the necessity of dimension-wise disentanglement for downstream tasks via extensive amounts of ablation studies. Through experiments, the authors examined 1) effects of attenuating disentanglement, 2) general-purpose vs disentangled training, 3) How different disentanglement/informativeness metrics correlate with downstream tasks, and 4) the correlation between LR and some disentanglement metrics. Every claim/reasoning is supported by experiments.

--The authors conduct multiple runs of experiments, and cover different kinds of model architectures and disentanglement training methods and metrics. The exploration is pretty thorough.

Weakness:

--The exploration should be expanded to other tasks and domains.

--There are a lot of general-purpose pre-training algorithms, but the authors mostly focus on BOYL.

--Though two-stage training is acceptable, what if the experiments are conducted in a joint-training setup where disentangling losses are used to regularize the supervised loss?


**Summary Of The Paper:**

This paper studies dimension-wise disentangled representations for downstream applications. Through extensive experiments, the authors conclude that disentanglement is not a necessity for achieving good performance in downstream tasks, and general-purpose representation learning methods could achieve better (or at least competitive performance) than disentanglement methods.

The authors show that Logistic regression accuracy on factor classification well-correlates with downstream task performance. The reason that we feel disentanglement is useful for downstream tasks is presumably due to the positive correlation between LR and disentanglement metrics.


**Summary Of The Review:**

Overall, I think the paper shows something interesting, and I’m inclined to recommend this paper.

1.     One thing I’m not sure about is how thoroughly that people have studied the importance of disentanglement for downstream tasks. This work is related to one previous ICLR submission (https://openreview.net/pdf?id=1JN7MepVDFv) where the authors studied the relationship between multi-task learning and disentanglement. The authors showed that disentanglement emerges naturally from MTL. The paper was rejected due to the main issue that MTL results in more extraction of information and that it is hard to disentanglement from the disentanglement metrics used. The authors (of the MTL work) also claimed that it is inconclusive whether disentangled representations have a clear positive impact on the model performance. In this work, the authors give a clearer explanation.

2.     Recent SSL framework also shows that a linear head on top of a pre-trained general-purpose encoder can achieve near-SOTA performance.

---

> ### Author Response · Authors · 2022-11-16
> **Response to Reviewer hRSM**
>
> Dear reviewer,
>
> We are very grateful for your reviews. Below are our responses to your questions.
>
> **Q1: The exploration should be expanded to other tasks and domains.**
>
> **A1:** Thanks for pointing this out!  Among the empirical evidence supporting the benefits of disentanglement [1,2,3], abstract visual reasoning is the task where disentanglement is found to be the most helpful with little compromise. Besides, abstract reasoning task is challenging and significant (see Section 2). As our purpose is to show that disentanglement is unnecessary, we use abstract reasoning as a benchmark given limited computational resources. We include more representation learning methods and downstream model structures to ensure the generalizability of our conclusions.
>
> We have provided a more detailed review of previous works to make our motivation clear in Section 2.
>
> **Q2: There are a lot of general-purpose pre-training algorithms, but the authors mostly focus on BOYL.**
>
> **A2:** Thanks for your suggestions! We have conducted experiments on another general-purpose learning model, SimSiam[4]. We have included the results of 72 SimSiams on 3DShapes with WReN as stage-2 models in Section B.  SimSiams achieve better performance than DisVAEs. As for the correlations, LR still correlates with the downstream accuracy most through all considered steps. Therefore, our conclusions still hold for SimSiam, i.e., disentanglement is not necessary.
>
> **Q3: What if the experiments are conducted in a joint-training setup where disentangling losses are used to regularize the supervised loss?**
>
> **A3:**  In practice,  joint training is often difficult since downstream task supervision is often too limited to train models from scratch. Therefore we need large-scale unsupervised data to learn good representations for easier downstream learning. So the two-staged task is a more realistic setting to verify the purported benefits of disentanglement.
>
> [1] Van Steenkiste, S., Locatello, F., Schmidhuber, J., & Bachem, O. (2019). Are disentangled representations helpful for abstract visual reasoning?. *Advances in Neural Information Processing Systems*, 32.
>
> [2] Dittadi, A., Träuble, F., Locatello, F., Wüthrich, M., Agrawal, V., Winther, O., ... & Schölkopf, B. (2021). On the Transfer of Disentangled Representations in Realistic Settings. In *International Conference on Learning Representations*.
>
> [3] Locatello, F., Abbati, G., Rainforth, T., Bauer, S., Schölkopf, B., & Bachem, O. (2019). On the fairness of disentangled representations. *Advances in Neural Information Processing Systems*, *32*.
>
> [4] Chen, X., & He, K. (2021). Exploring simple siamese representation learning. In *Proceedings of the* *IEEE**/CVF Conference on* *Computer Vision and Pattern Recognition* (pp. 15750-15758).

---

### Official Review · Reviewer_RvuP · 2022-10-25

**Confidence:** 2
**Correctness:** 3
**Technical Novelty And Significance:** 3
**Empirical Novelty And Significance:** 3
**Recommendation:** 6

**Clarity, Quality, Novelty And Reproducibility:**

The paper is well written. The experiment is very thorough. A researcher should be able to reproduce the results.

**Strength And Weaknesses:**

As I was reading the paper I was confused by these two statements:

"However, on the abstract visual reasoning task, we find that rotating disentangled representations, i.e., multiplying the representations by an orthonormal matrix, has no impact on sample efficiency and final accuracy."

-and-

"Our finding demonstrates that disentanglement does not affect the downstream learning trajectory, which is against the commonly believed usefulness of disentanglement. On the other hand, it is not surprising since we apply an invertible linear transform. We can observe that Logistic Regression (LR) accuracy remains 100% before and after rotation, indicating that a simple linear layer could eliminate the effects of rotation."

It appeared the authors believed that the multiplication would destroy the disentangled representation, and used this to prove their thesis. But, they later acknowledged that LR is capable of reversing the multiplication. I am at a loss for why to include this in the paper, and why to have it so early in the text.

The authors have a good experiment design, using learners that provide a disentangled representation (various VAE based approached) and compare against learned representations created by BYOL (which makes no claim to disentanglement). These representations are used to perform abstract visual reasoning tasks. They use the accuracy of using a logistic regression on the learned representation (informativeness) as a metric to show the usefulness of the representation.

**Summary Of The Paper:**

The authors call into question the conventional thinking that downstream tasks (the take a representation as input) benefit from a disentangled representation. Their results are that the informativeness of the representation not the disentangled nature of is what results in improved downstream performance.

**Summary Of The Review:**

The authors demonstrate for RPM downstream tasks the informativeness (the ability for the logistic regression to achieve a level of accuracy using the representation) is more indicative of downstream performance that the disentanglement of the representation. My biggest issue with the paper is their use of the orthonormal matrix multiplication to re-entangle the representation.

---

> ### Author Response · Authors · 2022-11-16
> **Response to Reviewer RvuP**
>
> Dear Reviewer,
>
> We appreciate your review. Below are our responses.
>
> **Q1:** **Why do we include orthonormal matrix multiplication?**
>
> **A1:** We include this to show that destroying disentanglement (by rotation) does not affect downstream performance. By saying LR can reserve such rotation early in the text, we aim to motivate our research questions. I.e., though rotation entangles representations, a simple LR could recover it, let alone the complex neural network we use for downstream tasks (WReNs and Transformers). We then empirically verify this in Figure 2. Starting from this, we challenge the necessity of disentanglement.

---

### Official Review · Reviewer_r3zU · 2022-10-27

**Confidence:** 5
**Correctness:** 2
**Technical Novelty And Significance:** 1
**Empirical Novelty And Significance:** 1
**Recommendation:** 3

**Clarity, Quality, Novelty And Reproducibility:**

**Clarity:** Good. (see strength above)

**Quality:** Poor.
- Main contribution of "challenging the necessity of disentangled representations" is questionable (see weaknesses above).
- Incorrect evaluation of sample efficiency (see weaknesses above).


**Novelty:** Poor/limited.
  - Many works have thoroughly investigated the correlation between disentanglement and downstream performance. Novelty of this work is unclear in relation to those works, except for the questionable focus on "necessity".

**Strength And Weaknesses:**

**Strengths:**
- **Clear writing:** this paper was easy to read and understand, communication was clear.
- **Important question/investigation:** The usefulness of disentanglement for downstream tasks is an open and important question.

**Weaknesses:**
- **Not clear who claimed disentanglement is *necessary* for downstream tasks:**
  - I agree with the authors that it is often said/believed that disentangled representations *can* be beneficial for downstream tasks.
  - However, throughout this paper, including in the title, the authors speak of the *necessity* of disentanglement representations for downstream tasks, claiming that they "challenge the necessity of disentanglement for downstream tasks". It is not clear to me who has claimed that disentanglement is *necessary*, i.e. why it should be surprising that there exists a downstream task for which disentanglement does not help---this seems like a trivial statement.
  - This is a major weakness since "challenging the necessity of disentanglement" is perhaps the central contribution of this paper.
  - A more interesting and non-trivial alternative question could be: when does disentanglement help, and when does it not? This is what prior works investigated [2,3,4], Naturally, there will be tasks for which it helps and tasks for which it does not.
- **Incorrect evaluation of sample efficiency:**
  - The authors use learning curves (gradient step vs. accuracy) to evaluate "sample efficiency". While each step sees new samples, this is fundamentally flawed since it convolutes sample efficiency (the performance with N samples) and update/step efficiency (the performance with M gradient updates).
  - This is a major weakness since sample efficiency is one of the most commonly-purported downstream benefits of disentanglement [1,2,3,4], making it central to the authors' claims.
- **Insufficient comparison to related studies on disentanglement and downstream performance:**
  - Many works have thoroughly investigated the correlation between disentanglement and downstream performance [2,3,4]. These works were much wider in their scope (tasks, datasets, representations) and reached different conclusions. Despite the attempt to undermine these studies in the related work and on page 9, I was left unconvinced that the results in this paper are novel or should underline/question those that came before. A better comparison and explanation would help, or perhaps a more specific claim could be made (e.g. relating to this one reasoning task with neural-net architectures).
- **Minor:**
  - *Rotation-of-factors issue is well-known and studied:* The authors claim to "find that rotating disentangled representations [...] has no impact on [...] final accuracy". This seems unsurprising given the well-known issue of rotation of factors in a linear factor-analysis model [5, sec. 9.6], which also leads to the condition in independent components analysis (ICA) that at most one of the factors can be Gaussian [6]. It was also discussed in [7]. Finally, note that it has also been investigated by a very recent (perhaps concurrent) work [8] which proposes a new notion of disentanglement that is unaffected by such rotations [8].
  - *The term "informativeness" is either overloaded or not cited:* The authors seem to use the term "informativeness" to refer to the linear-classifier performance in classifying the ground-truth factors from the learned (disentangled) representation. If so, this is precisely the definition of the "informativeness" metric in [7], but the authors never mention this relation. If this is in fact the same metric, the authors should appropriately cite, or if not, they should use a different name/term to avoid overloading an existing metric for evaluating disentangled representations.
  - *Unclear to me why disentanglement should help the specific abstract-reasoning task used:* Are only a subset of the factors needed? Do only a subset of the factors change? Is it surprising that disentanglement does not help?
  - *Incorrect and imprecise statements:*
    - In the second paragraph of the introduction, the authors claim that the abstract-reasoning task is general and widely-adopted, while other downstream evaluation tasks are "trivial or domain-specific", citing many past evaluations. I have to disagree with this presentation of abstract visual-reasoning as the holy-grail of downstream evaluations, and suggest that the wording be toned down.
    - *"Locatello et al. (2019b) proves their agreement on VAE methods"* -- Locatello et al. do not prove the agreement of different disentanglement metrics -- many of them measure different things.
   - *"it takes hours to develop the Gradient Boosting Trees required [to evaluate DCI disentanglement]"*: GBTs are not _required_ to evaluate DCI disentanglement---any classifier can be used, including those with a lower cost (e.g. random forests).

[1] Bengio, Y., Courville, A., \& Vincent, P. (2013). Representation learning: A review and new perspectives. _IEEE Transactions on Pattern Analysis and Machine Intelligence_, 35(8), 1798-1828.

[2] Locatello, F., Bauer, S., Lucic, M., Rätsch, G., Gelly, S., Schölkopf, B., \& Bachem, O. (2020). A Sober Look at the Unsupervised Learning of Disentangled Representations and their Evaluation. _Journal of Machine Learning Research_, 21, 1-62.

[3] Van Steenkiste, S., Locatello, F., Schmidhuber, J., \& Bachem, O. (2019). Are disentangled representations helpful for abstract visual reasoning?. _Advances in Neural Information Processing Systems_, 32.

[4] Dittadi, A., Träuble, F., Locatello, F., Wüthrich, M., Agrawal, V., Winther, O., ... \& Schölkopf, B. (2021). On the Transfer of Disentangled Representations in Realistic Settings. In _International Conference on Learning Representations_.

[5] Mardia, K. V., Kent, J. T., \& Bibby, J. M. (1979). _Multivariate Analysis_. Academic Press, London.

[6] Hyvarinen, A., Karhunen, J., \& Oja, E. (2001). _Independent Component Analysis_. Wiley.

[7] Eastwood, C., \& Williams, C. K. I. (2018). A framework for the quantitative evaluation of disentangled representations. In _International Conference on Learning Representations_.

[8] Eastwood, C., Nicolicioiu, A. L., von Kügelgen, J., Kekić, A., Träuble, F., Dittadi, A., \& Schölkopf, B. (2022). DCI-ES: An Extended Disentanglement Framework with Connections to Identifiability. _arXiv preprint arXiv:2210.00364_.

**Summary Of The Paper:**

This paper investigates the correlation between dimension-wise disentanglement scores and downstream performance. In particular, it does so when using MLPs or Transformers to perform the task of abstract visual reasoning using the learned representation. After observing a poor correlation on this task, it concludes that disentanglement is not *necessary* for good downstream performance.

**Summary Of The Review:**

While this paper is well written and explores a question that is both open and important, it ultimately fails to meet the standards required for acceptance. In particular: (1) its central claim---disentanglement is not *necessary* for downstream tasks---seems trivial; (2) its evaluation of sample efficiency is incorrect (this is one of the main purported benefits of disentanglement); and (3) its novelty/difference in comparison to prior such evaluations is not made sufficiently clear.

---

> ### Author Response · Authors · 2022-11-16
> **Response to Reviewer r3zU**
>
> Dear reviewer,
>
> Thank you so much for your review and suggestions. Below are our responses to your questions.
>
> **Q1:** **Not clear who claimed disentanglement is** ***necessary*** **for downstream tasks.**
>
> We would like to respond to your question by point.
>
> **Q1.1:** **The *necessity* claim.**
>
> **A1.1:**   [3] claimed the *necessity* of disentanglement for the abstract visual reasoning task.  [3] concluded that "disentangled representations *do in fact lead* *to* better down-stream performance". They show that disentanglement is the dominant factor determining downstream performance: the correlation of disentanglement exceeds other metrics by a considerable margin (see their Figure3 and Figure6). They claimed that "representations that are more disentangled give rise to better relative performance consistently throughout all phases of training."  Moreover, similar results are reached in other downstream tasks [2,4].
>
> **Q1.2: Why should it be surprising?**
>
> **A1.2:** Our results are against previous empirical evidence in [3]. We reach opposite conclusions on the same downstream task (abstract visual reasoning), representations (DisVAE's), downstream network (WReNs) training with similar steps (100K), and datasets (3DShapes and Abstract dSprites). Namely, [3] reported (1) high correlations between disentanglement and downstream accuracy, (2) nonexistent or weak correlations from informativeness, and (3) disentangled representations yield better downstream accuracy using relatively few samples. Our observations disagree with (1), (2), and (3). Moreover, our conclusions still hold for other representations (BYOL's) and downstream networks (Transformers).
>
> **Q1.3:** **A more interesting and non-trivial alternative question could be: *When does disentanglement help, and when does it not?***
>
> **A1.3:** We think it is significant to show disentanglement is not helpful in abstract reasoning.  The prevalent idea in the community is to verify the purported benefits of disentanglement in realistic tasks, and there is primarily positive evidence [2,3,4].  As we come to the opposite conclusion on the important and well-recognized abstract reasoning [3,10,11], we think our observations are worth studying and have considerable implications for the disentanglement community.
>
> We have made several contributions by challenging the necessity of disentanglement in abstract reasoning. We address previous studies' underestimation or ignorance of informativeness [2,3,4].  By this, we conclude the significance of informativeness. Further, we reveal that informativeness confounds the relationship between disentanglement and downstream performance.
>
> **Q2: Incorrect evaluation of sample efficiency.**
>
> **A2:** Both quantities measure sample efficiency: (1) convergence accuracy by training with different sizes of samples and (2) accuracy at different steps, with each step seeing fresh samples.  (2) is adopted by [3]. According to the task definition of abstract visual reasoning in [3], every step sees fresh samples. For better comparison, we follow [3] to use (2) as a measurement of sample efficiency.  We have added an explanation in Section 4.1.
>
> For completeness, we train the models in figures of learning curves (Figure 3 and Figure 4) with fixed sample sizes until convergence. We show the corresponding accuracy v.s. #samples curves in Figure 13 and Figure 14. We can observe that the rankings of the curves agree well when the horizon axis is the number of learning steps or sample size.
>
> **Q3: Insufficient comparison to related studies on disentanglement and downstream performance.**
>
> **A3:**  [3] is the most related to our study. We reach opposite conclusions against [3]'s on the same downstream task (abstract visual reasoning), representations (DisVAE's), downstream network (WReNs) training with similar steps (100K), and datasets (3DShapes and Abstract dSprites). This is because [3] yields falsely low correlations of informativeness due to underestimation and its study only constrains on DisVAEs.
>
> [2,4] investigates different tasks. And they also ignore the confounding of informativeness and are still limited to DisVAEs. Our study mainly focuses on the setting of abstract reasoning. Thus, [2,4]'s tasks are out of our scope.
>
> We have included comparisons with previous works in Section 2 in the rebuttal revision.

---

> > ### Author Response · Authors · 2022-11-16
> > **Response to Reviewer r3zU: Minors**
> >
> > **Minors**
> >
> > - *Rotation-of-factors issue is* *well-known and studied:*
> >   - It is surprising in the disentanglement community. Previous downstream studies don't realize rotation has no impact on downstream performance [2, 3, 4]. But it is well-known that disentangled representations being variant to rotation-of-factors is a corollary of disentanglement's definition (see Section 1 and 4.2). [7], [8] do include discussions on rotation, but their focus is *informativeness* rather than *disentanglement*. We acknowledge that this issue is deeply researched in some areas like ICA [5, 6]. But it is not the case in the disentanglement community.
> > - *The term "informativeness" is either overloaded or not cited*
> >   - We appreciate you for pointing this out. It is true that informativeness here is the same as in [7] (if measured by logistic regression). We have added a citation there.
> > - *Unclear why disentanglement should help the specific abstract-reasoning task used.*
> >   - Your understanding is correct. Only a subset of factors is set to be the reasoning attributes. Please refer to Section A.4 for the setting of the task.
> > - *Incorrect and imprecise statements*
> >   - Thanks for your suggestions! We have toned down our claim in Section1 and modified our claim on the relationship among metrics.
> > -  *GBTs are not required to evaluate DCI disentanglement*
> >     - We agree with you that DCI can use any classifier. Our study uses GBT to align with previous works [2,3,4,9] for better comparison. We have added an explanation in Section A.3.
> >
> > [1] Bengio, Y., Courville, A., & Vincent, P. (2013). Representation learning: A review and new perspectives. *IEEE* *Transactions on Pattern Analysis and Machine Intelligence*, 35(8), 1798-1828.
> >
> > [2] Locatello, F., Bauer, S., Lucic, M., Rätsch, G., Gelly, S., Schölkopf, B., & Bachem, O. (2020). A Sober Look at the Unsupervised Learning of Disentangled Representations and their Evaluation. *Journal of* *Machine Learning* *Research*, 21, 1-62.
> >
> > [3] Van Steenkiste, S., Locatello, F., Schmidhuber, J., & Bachem, O. (2019). Are disentangled representations helpful for abstract visual reasoning?. *Advances in Neural Information Processing Systems*, 32.
> >
> > [4] Dittadi, A., Träuble, F., Locatello, F., Wüthrich, M., Agrawal, V., Winther, O., ... & Schölkopf, B. (2021). On the Transfer of Disentangled Representations in Realistic Settings. In *International Conference on Learning Representations*.
> >
> > [5] Mardia, K. V., Kent, J. T., & Bibby, J. M. (1979). *Multivariate Analysis*. Academic Press, London.
> >
> > [6] Hyvarinen, A., Karhunen, J., & Oja, E. (2001). *Independent Component Analysis*. Wiley.
> >
> > [7] Eastwood, C., & Williams, C. K. I. (2018). A framework for the quantitative evaluation of disentangled representations. In *International Conference on Learning Representations*.
> >
> > [8] Eastwood, C., Nicolicioiu, A. L., von Kügelgen, J., Kekić, A., Träuble, F., Dittadi, A., & Schölkopf, B. (2022). DCI-ES: An Extended Disentanglement Framework with Connections to Identifiability. *arXiv preprint arXiv:2210.00364*.
> >
> > [9] Locatello, F., Abbati, G., Rainforth, T., Bauer, S., Schölkopf, B., & Bachem, O. (2019). On the fairness of disentangled representations. *Advances in Neural Information Processing Systems*, *32*.
> >
> > [10] Locatello, F., Poole, B., Rätsch, G., Schölkopf, B., Bachem, O., & Tschannen, M. (2020, November). Weakly-supervised disentanglement without compromises. In *International Conference on* *Machine Learning* (pp. 6348-6359). PMLR.
> >
> > [11] Schölkopf, B., Locatello, F., Bauer, S., Ke, N. R., Kalchbrenner, N., Goyal, A., & Bengio, Y. (2021). Toward causal representation learning. *Proceedings of the* *IEEE*, *109*(5), 612-634.

---

> > > ### Author Response · Authors · 2022-11-25
> > > **Do our responses address your concerns?**
> > >
> > > Dear Reviewer r3zU,
> > >
> > > We would like to ask if our responses have adequately addressed your concerns and answered your questions. If not, we are happy for any further discussions.
> > >
> > > Regards,
> > >
> > > The authors.

---

> > > > ### Comment · Reviewer_r3zU · 2022-11-25
> > > > **Response to authors**
> > > >
> > > > I would like to thank the authors for their response and engaging with my points/concerns/weaknesses. However, my main concerns regarding the claim of "necessity", the incorrect evaluation of sample efficiency, and the insufficient discussion of (and comparison to) related evaluations were not addressed. In particular, I disagree with the presented arguments/responses to these weaknesses, as detailed below. As a result, I retain my original score.
> > > >
> > > >
> > > > - **Necessity of disentanglement**: Despite your interpretation, none of the quotes you mention actually illustrate that [3] claimed that disentanglement was *necessary* for downstream tasks. In particular, I interpret both quotes *"disentangled representations do in fact lead to better down-stream performance"* and *"representations that are more disentangled give rise to better relative performance consistently throughout all phases of training"* as speaking of **general trends rather than absolute necessities**. From this interpretation, finding a single counter-example is not so surprising or interesting.
> > > >
> > > > - **Sample efficiency:** I disagree that performance after M steps is a good measure of sample efficiency, even if fresh samples are used. As in my original comment, this ultimately conflates sample efficiency (performance with N samples) with update/step efficiency (performance with M gradient updates, i.e. the speed of learning). Given this conflation, one cannot conclude that observed trends are due to sample efficiency---they may well be due to update efficiency (i.e. speed of learning).
> > > >
> > > > - **Related studies on disentanglement and downstream performance:** I appreciate the added paragraph in Section 2 explaining the relation to [3]. However:
> > > >   - it seems strange to me that all other disentanglement evaluations [2,4,A] are out-of-scope because they look at different tasks. I think they should still be discussed, but I guess this comes back to our disagreement on whether or not showing poor correlation on a single task is sufficient/interesting.
> > > >   - [A], which was not cited, already correct for the cofounding of informativeness and find similarly-poor correlations so should be cited and discussed here. In fact, the results of [A] draw into question the novelty of: (i) finding poor correlations; and (ii) adjusting for informativeness.
> > > >
> > > >
> > > > [A] Träuble, F., Dittadi, A., Wuthrich, M., Widmaier, F., Gehler, P. V., Winther, O., ... \& Bauer, S. (2022). The Role of Pretrained Representations for the OOD Generalization of RL Agents. In _International Conference on Learning Representations_.

---

> > > > > ### Author Response · Authors · 2022-11-27
> > > > > **Response to Reviewer r3zU**
> > > > >
> > > > > Dear reviewer,
> > > > >
> > > > > Many thanks for your detailed response. We would like to discuss your concerns by points.
> > > > >
> > > > > **Q1: Necessity of disentanglement**
> > > > >
> > > > > **A1:** We agree with you that [3] did not literally claim the necessity of disentanglement. However, even though no previous work has explicitly claimed it, we think our work contributes to answering whether disentanglement is necessary with extensive empirical evidence.
> > > > >
> > > > > [3] provided encouraging results that disentanglement benefits abstract reasoning. And previous works [3,4,9] advocated using disentangled representations to complete downstream tasks from the results of large-scale experiments. Though they have not straight revealed the necessity,  it is a natural question to ask: Given considerable positive evidence, is disentanglement necessary? No study has provided an answer. We step forward to study the necessity and provide evidence against the common appeal of disentangled representations.
> > > > >
> > > > > Thanks for providing more details of your concern about the *necessity* claim. We will rephrase our contributions in a newer version.
> > > > >
> > > > > **Q2: Sample efficiency**
> > > > >
> > > > > **A2:** We still insist that both measurements are valid with their equivalence and previous practice [3, 10, A].
> > > > >
> > > > > In the rebuttal revision, we have shown the equivalence of these two measurements in our setting. We train the models in figures of learning curves (Figures 3 and 4) with fixed sample sizes until convergence. We show the corresponding accuracy v.s. sample size curves in Figures 13 and 14. The rankings at a curtain step (sample size) are consistent between both types of curves.
> > > > >
> > > > > Our choice to use training curves (with fresh samples per step) has been adopted in previous works about disentanglement's downstream tasks [3, 10, A]. Moreover, it is more suitable for our task settings [3].
> > > > >
> > > > > **Q3: Related studies on disentanglement and downstream performance**
> > > > >
> > > > > **A3:** We appreciate for pointing out this. [2, 4, A] are out of our scope because we consider different task settings. According to [3, 4, 9], [2]'s task setting is trivial, i.e., recovering factors from representations, and [2] only studies sample efficiency but misses final performances. In addition, [4, A]  investigate specific OOD tasks, which are too sophisticated than our focused settings.  We are interested in more common and basic tasks.
> > > > >
> > > > > Our findings are surprising for the considered tasks. Though in the OOD settings, there is less encouraging evidence of the usefulness of disentanglement [4, A]. But it is not the case for more basic tasks. Without overly difficult generalization requirements, disentanglement is found to correlate highly with downstream performance [3]. As for informativeness, its correlation is lower or non-existent in [3], let alone confounding disentanglement and downstream performance.
> > > > >
> > > > > Thanks for your references. We would like to provide more details and comparisons of related studies and add a citation of [A] with discussion in a future revision.

---

### Author Response · Authors · 2022-11-16
**Rebuttal Revision**

Dear reviewers,

We appreciate the time and effort you have dedicated to providing valuable feedback.

We have been able to incorporate changes to reflect most of your suggestions. We have highlighted the changes in **blue** within the rebuttal revision.

## Changelog

1. We modify the presentation of the abstract reasoning task and rephrase our contributions in **Section 1**.
2. We add citations for *informativeness* in **Sections 1, 3, and 4**.
3. We provide more detailed comparisons with previous works in **Section 2**.
4. We add more explanations to the relationship of representation metrics in **Section 3**.
5. We give more illustration of our measurements of sample efficiency in **Section 4**. And we report the accuracy-#samples curves in **Figures 13 and 14 in Section C.2**.
6. We provide more results for **Table 1**:
   1. We add STDs for $\overline{\operatorname{WReN}}$ and $\overline{\operatorname{Trans.}}$ in **Table 1**.
   2. We report the model type and step achieving the performance in Table 1 in **Table 3 in Section C.1**.
7. We conduct new experiments on a new type of general-purpose model, SimSiam. The SimSiam-WReN results on *3DShapes* are in **Section B**.
8. We report the mean values and STDs for metric scores in **Table 4 in Section C.4**.

---

### Decision · Program_Chairs · 2023-01-20

**Decision:**

Reject

**Justification For Why Not Higher Score:**

There are fundamental flaws in the main experiments of this paper.

**Justification For Why Not Lower Score:**

N/A

**Metareview: Summary, Strengths And Weaknesses:**

This paper tries to investigate the correlation between disentanglement scores of representation and downstream performance on visual abstract reasoning tasks. While the paper investigates an important topic, it has fundamental flaws that lead me to recommend rejecting this paper:

1) The experiment with rotated representations provides no real insight for two reasons: The Stage 2 model used in the paper applies (learned) linear transformations to the representations that are usually initialized to random matrices. As a result, it appears obvious that adding an additional random perturbation would not impact the final performance. Furthermore, the rotation-of-factors issue is well-known and studied in the literature, and seems to be more of an issue of how disentanglement should be defined than whether disentanglement is useful for the considered task.

2) The second experiment that compares BYOL and DisVAE also has a fundamental flaw: The authors do not correctly measure sample efficiency (performance at a certain number of gradient steps vs performance with a certain number of samples). This is highly problematic as sample efficiency is one of the commonly-purported downstream benefits of disentanglement.

3) The authors claim that "dimension-wise disentangled representations are not necessary for downstream tasks using neural networks that take learned representations as input." in a paper with the title "ON THE NECESSITY OF DISENTANGLED REPRESENTATIONS FOR DOWNSTREAM TASKS". However, the paper only looks at one type of downstream task (abstract visual reasoning). Given the limited tasks and if the results were sound, it seems like a strong overgeneralization of the results to claim that "disentangled representations are not necessary for downstream tasks using neural networks that take learned representations as input".